# Genetic barcoding uncovers the clonal makeup of solid and liquid biopsies and their ability to capture intra-tumoral heterogeneity

Antonin Serrano [1,2,3,4,5,6,16✉], Tom S Weber [3,4,16✉], Jean Berthelet[1,2], Sarah Ftouni[5,7,8], Farrah El-Saafin [1,2], Samuel Lee[1,2], Elgene Lim[9,10,11], Emmanuelle Charafe-Jauffret[12], Christophe Ginestier [12], David Williams[1,2,13,14], Frédéric Hollande [5,6], Belinda Yeo [1,2,15], Sarah-Jane Dawson[5,7,8,17], Shalin H Naik[3,4,17] & Delphine Merino [1,2,3,4,17✉]

## Abstract

Intratumoral heterogeneity (ITH) is fueling tumor progression in breast cancer, as specific clones present within a tumor may have a selective advantage to colonize distant organs and escape therapy. Accurate sampling of ITH is therefore a pressing challenge in clinical oncology to adequately predict recurrence and inform rational and personalized therapies. Here, we used genetic barcoding to track the spatiotemporal composition of human breast cancer clones in six preclinical models—across two cell lines and four patient-derived xenografts (PDXs). This allowed a direct side-by-side quantitative comparison of both intra-tumor clonal composition and how that composition was reflected in needle biopsies and cell-free DNA (cfDNA). These analyses highlighted several biologically and clinically relevant findings. First, the use of barcoding revealed that clonal diversity in the center of non-necrotic primary tumors was significantly higher than in the periphery. Second, cfDNA barcode analysis suggested that DNA 'shedding' in the vasculature varied widely, not only depending on necrosis and tumor burden but also across models. Third, combining information captured in both solid and liquid biopsies can provide a more robust assessment of tumor clonal composition. Taken together, these results showcase the utility of these barcoded models to optimize the use of solid and liquid biopsies as surrogates of tumor heterogeneity.

**Keywords** Breast Cancer; Biopsies; Barcoding; cfDNA; Tumor Heterogeneity
**Subject Category** Cancer

## Introduction

Breast tumors are composed of a complex patchwork of cancer clones, each with distinct abilities to invade locally and distally (Aleckovic et al, 2019). While the exact mechanisms underpinning metastasis progression and drug resistance are not yet fully elucidated, intra-tumoral heterogeneity (ITH), including genetic (Turajlic et al, 2019) (DNA mutations) and non-genetic (Hinohara and Polyak, 2019) (e.g., epigenetic and transcriptomic) heterogeneity, correlates with disease progression (Marusyk et al, 2020; Yang et al, 2017). However, the contribution of individual clones to disease progression is not necessarily correlated with their size in the primary tumor, as low-frequency clones can be highly metastatic in distant organs (Dagogo-Jack and Shaw, 2018; Hu et al, 2020b; Kim et al, 2018; Merino et al, 2019). These observations highlight the need to comprehensively assess clonal heterogeneity in primary tumors to better predict and treat recurrence.

Currently, breast cancer diagnosis and prognosis are based on the analysis of solid biopsies and specimens collected during surgery. The status of estrogen receptor, progesterone receptor, and overexpression of the human epidermal growth factor receptor 2 (HER2), as well as markers of aggressiveness, are routinely used to identify the subtype and grade of the disease, which in turn guides

[1]Olivia Newton-John Cancer Research Institute, Heidelberg, VIC 3084, Australia. [2]School of Cancer Medicine, La Trobe University, Bundoora, VIC 3086, Australia. [3]Immunology Division, The Walter and Eliza Hall Institute of Medical Research, Parkville, VIC 3052, Australia. [4]Department of Medical Biology, The Faculty of Medicine, Dentistry and Health Science, The University of Melbourne, Parkville, VIC 3010, Australia. [5]Collaborative Centre for Genomic Cancer Medicine, University of Melbourne, Melbourne, VIC 3010, Australia. [6]Plasticity, Heterogeneity and Tumour Microenvironment International Research Laboratory – PHANTOM - CNRS and The University of Melbourne Department of Clinical Pathology, University of Melbourne, Melbourne, VIC 3010, Australia. [7]Peter MacCallum Cancer Centre, Melbourne, VIC 3000, Australia. [8]Sir Peter MacCallum Department of Oncology, The University of Melbourne, Melbourne, VIC 3000, Australia. [9]Garvan Institute of Medical Research, Darlinghurst, NSW 2010, Australia. [10]St Vincent's Clinical School, Faculty of Medicine, UNSW Sydney, Darlinghurst, NSW 2010, Australia. [11]St Vincent's Hospital, Darlinghurst, NSW 2010, Australia. [12]CRCM, Inserm, CNRS, Institut Paoli-Calmettes, Aix-Marseille University, Epithelial Stem Cells and Cancer Laboratory, Equipe labellisée LIGUE contre le cancer, Marseille, France. [13]Department of Pathology, Austin Health, Heidelberg, VIC 3084, Australia. [14]Department of Clinical Pathology, The University of Melbourne, Parkville, VIC 3010, Australia. [15]Austin Health, Heidelberg, VIC 3084, Australia. [16]These authors contributed equally: Antonin Serrano, Tom S Weber. [17]These authors contributed equally as senior authors: Sarah-Jane Dawson, Shalin H Naik, Delphine Merino. ✉E-mail: antonin.serrano@unimelb.edu.au; weber.ts@wehi.edu.au; delphine.merino@onjcri.org.au

therapeutic decisions (Sorlie, 2004). However, the level of heterogeneity captured in biopsies is still unclear. Yet, since most efforts in precision oncology are focused on identifying biomarkers predictive of outcomes and drug response for individual patients (Fremd et al, 2019; Vagia et al, 2020), biopsy methods must provide a precise representation of the tumor complexity. Thus, a full appraisal of the different types of biopsies and how accurately they reflect tumor heterogeneity and predict therapeutic outcomes is required in the design of personalized therapies.

Liquid biopsies from blood specimens have emerged as a powerful option to detect and monitor tumor progression and treatment responses in cancer patients, based on the analysis of cell-free DNA (cfDNA) and circulating tumor DNA (ctDNA) (Alix-Panabieres and Pantel, 2021; Dawson et al, 2013; Fernandez-Garcia et al, 2019; Shaw et al, 2017). Previous studies indicated that ctDNA can capture the genomic ITH of patient tumors (Bettegowda et al, 2014), whereas core needle biopsies or fine-needle aspiration may not be entirely representative of the genomic landscape of tumors (Gerlinger et al, 2012; Russo et al, 2016; Yates et al, 2015). Furthermore, ctDNA biopsies have the advantage of being less invasive than solid biopsies and are more likely to reflect the overall heterogeneity of the disease when multiple lesions cannot be easily biopsied simultaneously in clinical settings (Lee et al, 2020).

While differences between biopsy methods have previously been investigated (Ding et al, 2019; Finzel et al, 2018; Russo et al, 2016), many studies focused on common driver mutations (e.g., TP53, MEK1, KRAS, EGFR) rather than characterizing overall clonal heterogeneity of the disease. A comparison of ctDNA with needle biopsies in gastrointestinal cancer indicated that ctDNA captured resistance alterations not found in matched tumor biopsies in 78% of cases (Parikh et al, 2019). This suggests that ctDNA may be a better surrogate for genomic heterogeneity than solid biopsies. However, the ability of cancer clones to shed ctDNA into the bloodstream might depend on the level of tumor necrosis (Cho et al, 2020) and on intrinsic characteristics of the clones (Zhou et al, 2019). Furthermore, the recent development of single-cell RNA sequencing (scRNA-seq) analysis and drug prediction from transcriptional signatures, rather than mutational status, highlights the utility of capturing and analyzing intact cells in cancer research and precision medicine (e.g., (Pellecchia et al, 2023; Van de Sande et al, 2023)). This is particularly relevant for non-genetic cancer alterations. Therefore, improvements in the collection of fresh tissues from needle biopsies would be clinically useful in diagnostics and personalized medicine.

Cellular barcoding strategies using genetic barcoding (Echeverria et al, 2018; Eirew et al, 2015; Merino et al, 2019; Wagenblast et al, 2015) or optical barcoding (Berthelet et al, 2021; Lewis et al, 2021) have emerged as powerful tools to study the heterogeneity of breast cancer primary tumors and metastases. These lentiviral-based labeling strategies allow the labeling of individual cancer cells with unique genetic or optical tags, respectively (Gui and Bivona, 2022; Howland and Brock, 2023; Serrano et al, 2022). As these tags are stably integrated into the genome of the transduced cells and transmitted to their progeny, clonal fate can be monitored in large populations of cells, regardless of the molecular profiles of the clones and their spatiotemporal evolution. These preclinical models enable a comprehensive analysis of disease clonality in multiple mice, tissues, and conditions (for instance, in the absence of

treatment, at multiple time points), and therefore provide quantitative and qualitative information that can't be easily captured using patient samples. Here, we leveraged the use of DNA-based cellular barcoding to label six human breast cancer xenograft models and explore the clonal repertoire captured in solid and liquid biopsies. By comparing the barcode repertoire in ex vivo needle biopsy samples and in the plasma of tumor-bearing mice, we tested the hypothesis that specific clones might be over-represented in solid biopsies, given their spatial distribution within primary tumors, whereas liquid biopsies might provide a more representative overview of clonal diversity. Furthermore, we explored factors influencing barcode detection in solid and liquid biopsies, such as tumor burden, tumor necrosis, and clonal identity. This analysis provided a unique opportunity to comprehensively assess the ability of different sampling methods to capture ITH.

## Results

### Clonal density is higher in the center of primary tumors

Previous studies using multi-region sequencing of patient samples have shown that cancer clones grow as patches within primary tumors (de Bruin et al, 2014; Gerlinger et al, 2012; Mo et al, 2024; Navin et al, 2011; Shah et al, 2012; Yates et al, 2015). To better understand the spatial distribution of the clones in primary tumors and how it affects the diversity captured in solid biopsies, a previously established simulation model was adapted (Waclaw et al, 2015) (Fig. 1A). This stochastic agent-based model predicts the 3D growth of cancer cells based on three cellular parameters: birth rate, death rate, and mobility. Mimicking the number of clones detected in previous cellular barcoding experiments (Merino et al, 2019), simulations of fat pad transplantation experiments were initiated in silico with a starting number of two hundred cells, where each cell was given a unique identity (using a virtual tag stored in the genotype slot), thereafter inherited by their respective progeny. No other change was made to the model, previously detailed in (Waclaw et al, 2015). As expected, the resulting simulation of clonal growth confirmed the clonal 'patchiness' of the tumors previously observed using genetic barcoding (Merino et al, 2019). However, the analysis of the spatial distribution of clones in three dimensions revealed that clonal density in the tumor center was higher than in the periphery (Fig. 1B). Indeed, virtual dissection of these tumors into five pieces and quantification of the number of clones in each piece indicated that the center of the tumor (piece E) exhibited a significantly increased number of clones. This heterogeneous topographical distribution in clonal density observed throughout the simulated tumors raised an interesting question. Namely, does this hold true in an in vivo situation, where additional factors like mechanical pressure and constraints imposed on a tumor growing in a mammary fat pad could influence clonal distribution?

To investigate this further, genetic barcoding was used to study the growth of cancer clones in vivo. In this context, cancer clones were defined by their barcode lineage rather than their molecular characteristics. Breast cancer cells from the MDA-MB-231 cell line were infected with a lentivirus pool containing ~2600 unique genetic tags at low multiplicity of infection. Infected cells were then sorted as GFP$^+$ and transplanted into the mammary fat pad of NSG (NOD/SCID/IL2Rγ$^{-/-}$) mice, as this model and strategy were

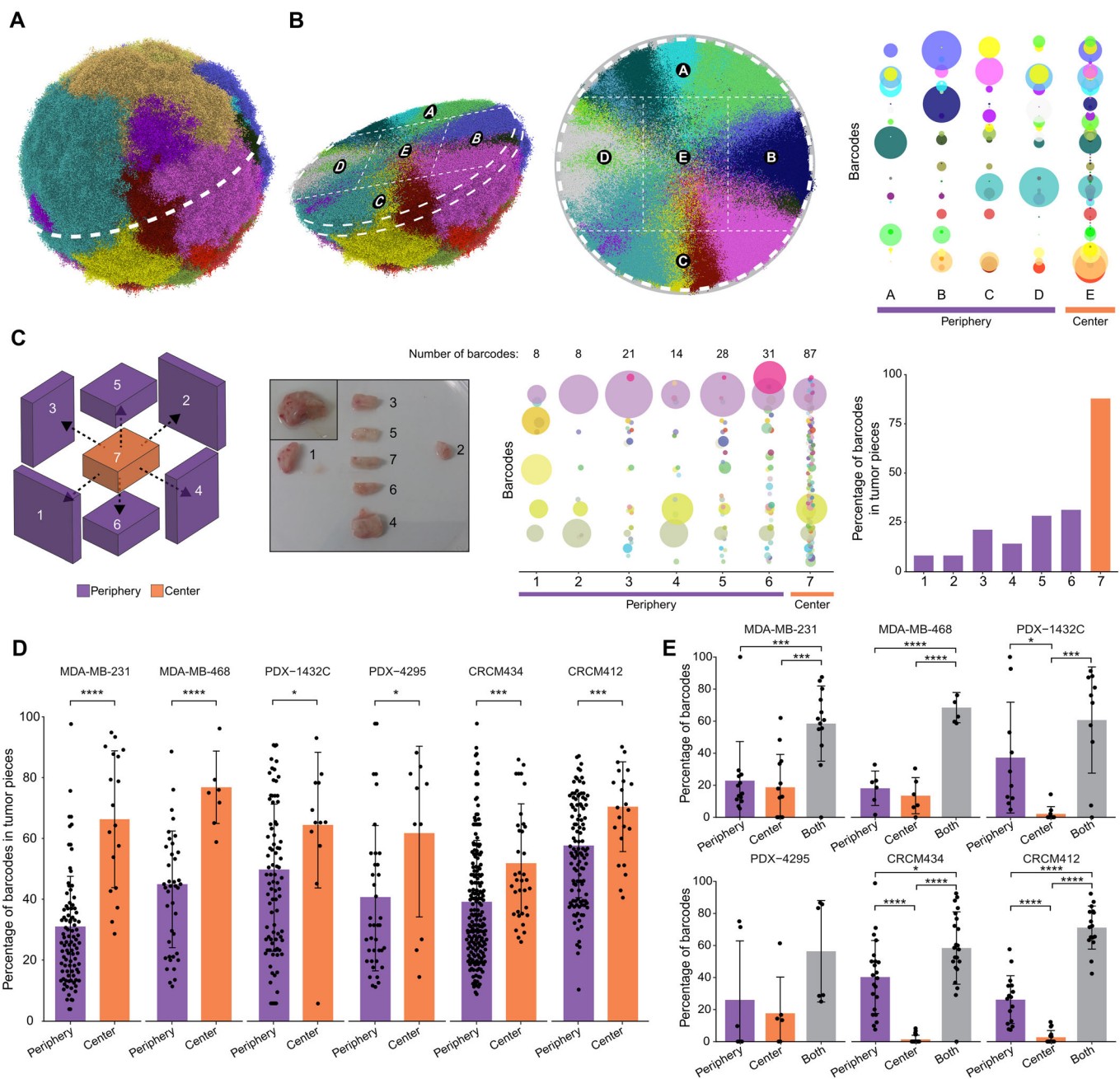

**Figure 1. Spatial heterogeneity in primary tumors.**

(A) 3D visualization of clonal growth by simulation, based on virtually barcoded cancer cells. (B) Dissection of the 3D virtual tumor into pieces (left), and representation of the barcode distribution in each piece (right). Each dot represents a barcoded clone. Its size correlates with the barcode frequency in each piece. (C) Dissection of an MDA-MB-231 primary tumor barcoded with genetic tags. From left to right: schematic overview of the dissection to isolate pieces from the center and periphery; bubble plot representing the clonal composition of each piece, and quantification of the percentage of barcodes present in each piece. Purple bars represent peripheral pieces, and the orange bar represents the center. (D) Percentage of barcodes detected in each piece, from the periphery (purple) and center (orange) of tumors, in multiple models. Each dot corresponds to a piece of primary tumor (on average, 1–2 pieces from the center and 6–12 pieces from the periphery for each tumor). Student's unpaired $t$ test, $P$ values: MDA-MB-231 = 2.4e-06, MDA-MD-468 = 4.1e-05, PDX-1432C = 0.0154, PDX-4295 = 0.0426, CRCM434 = 0.0003, CRCM412 = 0.0009. (E) Percentage of barcodes uniquely detected in the periphery (purple), tumor center (orange), or both (gray) of the tumor. Each dot corresponds to an individual tumor. Significance from one-way ANOVA followed by Tukey multiple comparisons test. $P$ values: MDA-MB-231 Periphery-Both=0.0009, Core-Both=0.0002; MDA-MB-468 Periphery-Both=1.6e-06, Core-Both=5.1e-07; PDX-1432C Periphery-Center=0.0229, Center-Both=0.0002; CRCM434 Periphery-Center=7.3e-09, Center-Both=1.5e-11, Periphery-Both=0.0053; CRCM412 Periphery-Center=3.4e-06, Center-Both=2.4e-13, Periphery-Both=4.1e-13. (D, E) ns non-significant, $P$ value > 0.05, *$P$ value < 0.05, **$P$ value < 0.01, ***$P$ value < 0.001, ****$P$ value < 0.0001. Error bars represent the standard deviation (SD) of the mean. MDA-MB-231, three independent experiments, $n = 5$, 4, 4, MDA-MB-468 one experiment, $n = 6$, PDX-1432C two independent experiments, $n = 6$, 4, PDX-4295 one experiment, $n = 6$, CRCM434 four independent experiments, $n = 7$, 6, 4, 5, CRCM412 four independent experiments, $n = 4$, 2, 5, 5. Source data are available online for this figure.

shown to produce primary tumors with multiple barcodes (Serrano et al, 2023). Tumors of 800 mm³ were harvested and cut into pieces of equal volume (Fig. 1C). Analysis of the barcode repertoire of tumor pieces after lysis, PCR amplification, and sequencing of the barcodes showed that each piece was unique in its barcode composition, with adjacent pieces displaying non-identical barcode repertoires, as previously described in other triple negative breast cancer (TNBC) models (Merino et al, 2019). Interestingly, and in agreement with the simulation in Fig. 1B, the results confirmed that pieces corresponding to the tumor center (Piece #7) contained, on average, approximately three times more distinct barcodes than pieces from the periphery (Fig. 1C).

To assess whether this observation was reproducible, we analyzed the tumors of multiple mice using barcoded MDA-MB-231 and MDA-MB-468 cell lines (Figs. 1D and EV1A–E; Dataset EV1). On average, the center of MDA-MB-231 and MDA-MB-468-derived primary tumors contained a large proportion of the total barcodes (66% and 77% for MDA-231 and MDA-468, respectively, Fig. 1D), compared to peripheral pieces. While the number of pieces analyzed at the center and periphery differed depending on the size and shape of the tumor, we confirmed that the higher proportion of clones in the center was not due to varying weight or volume of the tumor pieces (Fig. EV1F).

To determine whether enhanced clonal diversity in the center of the tumor could be generalized to more clinically relevant PDX models, the same experiments were performed with four different TNBC PDX models, generated from drug-naïve patient tumors. We confirmed that the receptor status of the original patient tumor was maintained in PDXs, as indicated by pathology reports (Fig. EV2A). Analysis of the barcode repertoire in primary tumors suggests that the number of barcodes varied between models (Fig. EV1A). However, when considering the Shannon diversity index, a measure that reflects both barcode richness and evenness, all PDXs, except PDX-4295, showed a higher diversity than at least one cell line model (Fig. EV1B). Furthermore, barcode analysis demonstrated that pieces from the center of the tumors contain a higher frequency of barcodes than those from the periphery in all PDXs, recapitulating the observation made in cell line xenografts (Figs. 1D and EV2B). The analysis of the Shannon diversity index in pieces from the center versus the periphery showed an increase of diversity in the center in the MDA-MB-231 model, but not in the other models (Fig. EV2C). This could be explained by the fact that clones were overall more evenly distributed in the periphery (Fig. 1C, bubble plot). Interestingly, PDX-1432C, a highly necrotic tumor both in mouse xenografts (Fig. EV2D) and in the original patient samples (Fig. EV2A), showed less difference in barcode number between the center and the periphery, compared to other models (Fig. 1D). In this case, it is possible that the necrosis resulted in the loss of many cells (and therefore barcodes) in the center of the tumor.

We next determined whether specific clones were more likely to be found in particular localizations, for instance, the center or periphery of primary tumors, or in the lungs. Overall, most clones were found in both the center and the periphery of primary tumors (Fig. 1E), and all clones detected in the whole lungs at high frequency were detected in both—center and periphery (Fig. EV3A, orange dots). This was due to the fact that, while clones were growing as patches, they were likely to be detected in multiple tumor pieces. Indeed, dominant clones (here defined as a clone

with a read frequency greater than 1% of the total reads of the tumor) were present in a larger number of pieces compared to minor clones, and the barcodes present exclusively in the center or the periphery were minor clones (Fig. EV3B,C). Altogether, in silico modeling and empirical results based on barcoding analysis converged towards the conclusion that in preclinical models, the tumor center is higher in clonal density compared to the periphery. This observation supports previous studies using the lentiviral gene ontology (LeGO) technology (van der Heijden et al, 2019), sequencing and modeling analyses (e.g., Chkhaidze et al, 2019; Lewinsohn et al, 2023; Noble et al, 2022), which have demonstrated that clones from the outer region of tumors are likely to have a higher growth rate compared to clones present in the center.

## The content of multiple needle biopsies from a given tumor is highly variable, but is likely to contain dominant clones

Solid biopsies are routinely used for the diagnosis and treatment of breast cancer patients. As our results in barcoded models confirmed the uneven distribution of the clones across the tumor, consistent with what has been previously observed in PDXs (Merino et al, 2019) and patient samples (Gerlinger et al, 2012; Mo et al, 2024; Yates et al, 2015), we next interrogated the level of heterogeneity captured by needle sampling in these barcoded models. To do so, multiple needle samplings were taken from primary tumors (Fig. 2A). Needles were directed towards the tumor center in four directions, and the barcode repertoire captured in the ex vivo biopsies was analyzed and compared to that of the whole tumor. In general, and as expected, the barcode frequency detected in one needle correlated with the barcode frequency in the primary tumors, as dominant barcodes in tumors were likely to be dominant in solid biopsies (Fig. 2B,C). However, several dominant clones in the primary tumors were not captured in needle biopsies, and a needle biopsy reaching the center of the tumor rarely reflected the full barcode heterogeneity of the primary tumor.

When comparing multiple biopsies from the same tumors, we found that the barcode repertoire varied depending on the orientation of the needle (Fig. 2C), with several minor and dominant barcodes not shared between biopsies collected in opposite directions (Figs. 2D and EV4A,B), likely due the variations in the spatial distribution of the clones in the tumor (Fig. 1B,C).

To determine whether solid biopsies were, overall, representative of the heterogeneity of primary tumors, we determined the percentage of reads from the primary tumors that were covered by barcodes detected in the needles. We found that clones detected in needle biopsies represented 80–90% of the tumor biomass (referring here to the total number of reads in the whole tumor), regardless of their orientation (Fig. EV4C).

Overall, these results demonstrated that, although the content of needle biopsies may vary depending on their orientation and dominant clones might be missed during sampling, these biopsies are largely representative of the primary tumor biomass.

## Barcode detection in cfDNA depends on the tumor burden and model

While the clonal repertoire captured in solid biopsies was biased by the spatial distribution of the clones within the tumor, we

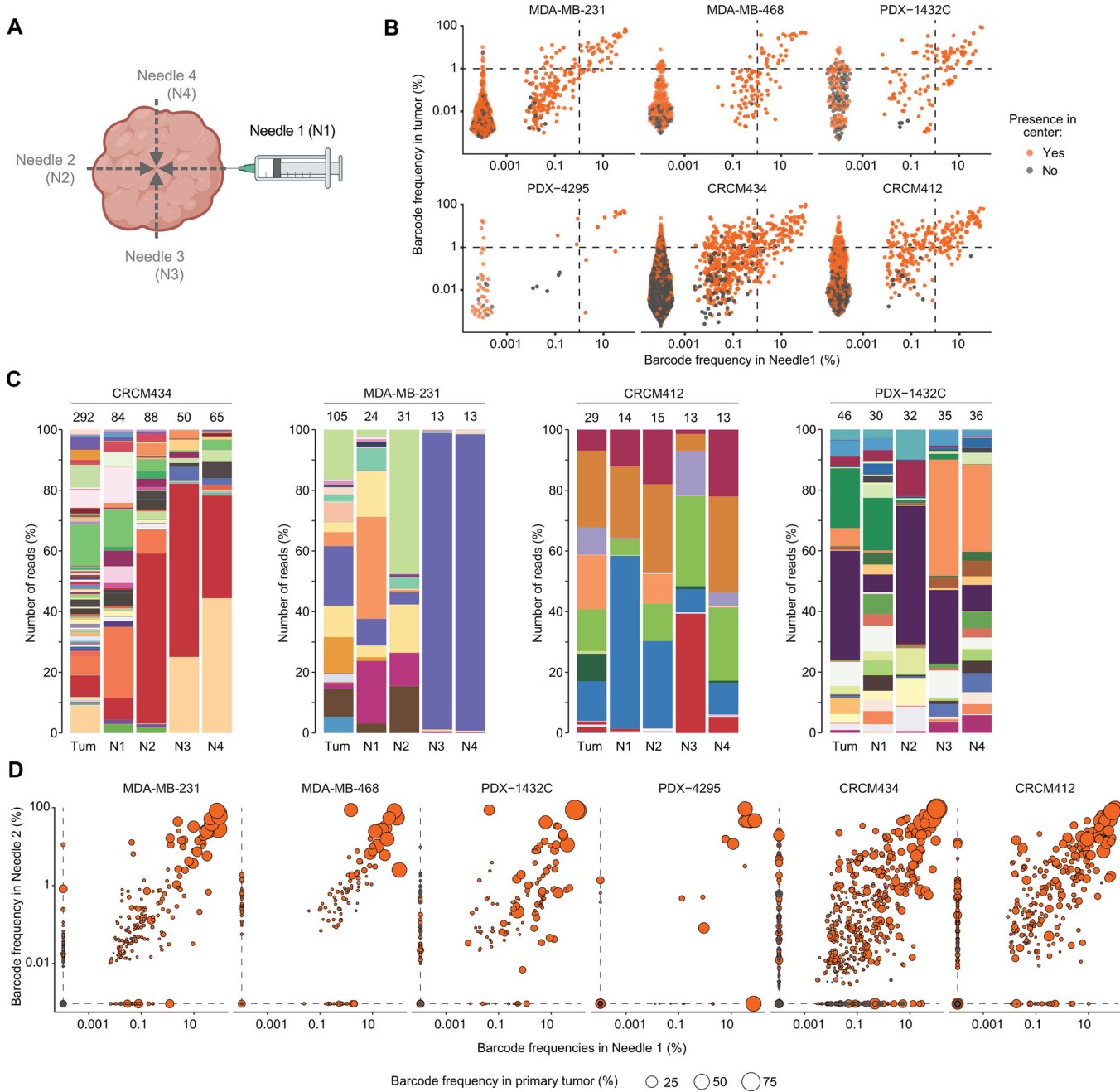

**Figure 2. Solid biopsies: clonal composition.**

(A) Schematic overview of needle sampling in a primary tumor. (B) Clonal relationship between the frequency of barcodes detected in the primary tumor and the needle sample N1 for each model. Each dot represents a barcode. Barcodes present in the center are represented by orange dots, and barcodes exclusively detected in the periphery are represented by gray dots. The dashed lines are drawn at 1% of barcode frequency. (C) Examples of clonal composition of primary tumor (left column) and each needle sample (N1 to N4) for four different models represented as a stacked histogram. Each color corresponds to a barcode. The number of barcodes per sample is indicated at the top of the histogram. (D) Relationship between barcodes detected in needle sample N2 and needle sample N1. Each dot represents a barcode; the size of the dot correlates with the frequency of the barcode in the primary tumor. Barcodes present in the tumor center are represented by orange dots, and barcodes exclusively present in the periphery are represented by gray dots. Dashed lines represent a barcode frequency of 0%. (B, D) MDA-MB-231, three independent experiments, $n = 1, 4, 4$, MDA-MB-468 one experiment, $n = 6$, PDX-1432C two independent experiments, $n = 4, 4$, PDX-4295 one experiment, $n = 4$, CRCM434, four independent experiments, $n = 7, 5, 3, 4$, CRCM412, four independent experiments, $n = 4, 2, 5, 5$. Source data are available online for this figure.

hypothesized that liquid biopsies, and in particular cfDNA isolated from the blood, may overall reflect the clonal makeup of the primary tumor.

To test this hypothesis, we assessed the presence of DNA barcodes in the cfDNA of tumor-bearing mice. We collected blood via the tail vein at different time points, when tumors reached 100 mm$^3$ (cfDNA1), 300 mm$^3$ (cfDNA2), or via heart terminal end bleed when the tumors reached 800 mm$^3$ (cfDNA3). After plasma separation, DNA was extracted and barcodes amplified by PCR run in five replicates to increase the probability of barcode recovery. The results indicated that barcode recovery in cfDNA depended on the models, even when the primary tumors were similar in size (Fig. 3A). For example, PDX-4295 and CRCM412 showed a higher rate of barcode recovery in plasma compared to other models, even when the tumors were small. This suggested that the onset of cfDNA shedding might vary between tumors, depending on the intrinsic properties of cancer cells.

Furthermore, within a given model, barcodes were not consistently detected in the plasma of all mice (Fig. 3A), even though the sequencing coverage was similar across cfDNA and other samples (Fig. EV5A). However, as expected, barcode recovery seemed to increase with tumor burden (Fig. 3A,B). At the last time point (cfDNA3), the larger volume of blood collected and the type of bleeding (heart bleed rather than tail vein, see methods) could have contributed to the increase in barcode detection. However, the analysis of cfDNA2 showed a higher recovery rate than cfDNA1 in several models, using the same sampling method, confirming that tumor size likely influenced DNA shedding.

Where possible, we compared the identity of the barcodes detected in the plasma over time (Fig. 3B). Surprisingly, we found that different barcodes were detected at various times in the same mice. The barcodes detected in cfDNA at early bleeding didn't necessarily reflect the full heterogeneity of the primary tumor analyzed at ethical endpoint. On the contrary, cfDNA collected at the last time point (cfDNA3) contained a barcode repertoire representative of the primary tumor (Figs. 3B,C and EV5B). These differences may be attributed to the amount of cfDNA present in blood, which likely depends on tumor burden, and the level of detection of DNA barcodes in these models. Even though barcode recovery from cfDNA significantly differed between models (Fig. 3A), when detected, only a small percentage of the barcodes from the primary tumor were represented, except for PDX CRCM412 (Figs. 3D and EV5C), and some of the dominant barcodes in the primary tumors were not detected in the plasma (Fig. 3B,C). However, clones that were detected in cfDNA contributed a significant percentage of the tumor biomass (Figs. 3E and EV5B, up to ~80% for PDX CRCM412), representing clones present in the center and periphery of the tumors (Fig. EV5D). Furthermore, cfDNA seemed to be a good surrogate of the heterogeneity captured in lungs, as highlighted in Fig. 3C. Overall, the barcodes captured in cfDNA and needle biopsies represented a similar proportion of the tumor biomass detected in lungs (Fig. EV5E).

Because the barcode repertoire detected in blood varied over time and between mice (Fig. 3A,B), we investigated whether this was due to differences in the clonal composition of the primary tumor. To do so, the same clones from PDX-1432C were implanted into multiple mice after in vivo expansion. These "sister" tumors, originating from the same clonal pool, were then analyzed when they reached specific sizes (Fig. EV5F), and the same volume of blood was collected for each mouse via heart bleed. Barcodes were then analyzed in primary tumors and plasma (Fig. EV5F). When barcode recovery was compared across multiple mice within these cohorts, we confirmed that, as shown in Fig. 3A, the recovery rate of barcodes in the plasma depended on the size of the primary tumor (Fig. EV5F). When considering barcode identity, the clonal repertoire of sister tumors was strikingly conserved over time, in each experiment (Fig. EV5G). Barcode recovery and frequency could differ from one mouse to another (Fig. EV5G). These differences could be due to the low level of cfDNA detected in these models or variations in cfDNA shedding depending on tumor vascularization. However, the results between mice were overall reproducible when the mice had similar tumors (Fig. EV5G), in contrast to cohorts where mice received different-pools of clones (Fig. 3A, PDX-1432C), suggesting that the nature of the clones may influence ctDNA shedding within a given model.

## cfDNA and solid biopsies can provide complementary information

As both solid and liquid biopsies captured varying degrees of heterogeneity, we compared the clonal repertoires obtained with each sampling technique in the same mice across the 6 xenograft models. Overall, taking into consideration cases where cfDNA was not detected, we found that solid biopsies reaching the center of the tumor or to half the radius captured a larger percentage of the primary tumor barcodes than cfDNA (Fig. 4A,B). Solid biopsy samples also showed higher diversity than cfDNA samples in most models, except for CRCM412 and PDX-4295 (Fig. EV6A). However, for those mice with barcodes detected in the plasma (limiting the number of samples analyzed to $n = 2$ for the MDA-MB-231 model and $n = 1$ for MDA-MB-468, Dataset EV1), the Pearson correlation between needle sampling or cfDNA and primary tumor showed limited differences (Fig. EV6B), and biopsies comprised both minor and dominant clones present in primary tumors (Fig. EV6C). It is important to note that in these preclinical models, the barcode content of biopsies varied significantly, not only between models, but also between mice from the same model, regardless of the method. This could be due to the low detection rate and resulting stochasticity obtained when sampling small amounts of tissue and limited blood volumes. Indeed, comparing the representativity of barcodes captured by needle and liquid biopsies in individual mice from the same cohort, three distinct patterns were identified. In the first pattern (for instance, Fig. 4C), the barcode repertoire in each biopsy sample was unique, as needle and liquid biopsies contained different sets of barcodes, whereas the cfDNA barcode repertoire better recapitulated tumor heterogeneity. In the second pattern (Fig. 4D), needle and liquid biopsies also showed distinct sets of barcodes, but solid biopsies were better surrogates of primary tumor ITH, while cfDNA captured only one clone from the primary tumor. The last pattern (Fig. 4E) showed similarity between needle and cfDNA biopsies, with both methods accurately representing the clonal heterogeneity of primary tumors.

The tumor biomass captured with each method was then quantified (Fig. 4F). Needle sampling was a better compendium of primary tumor heterogeneity compared to cfDNA (when detected

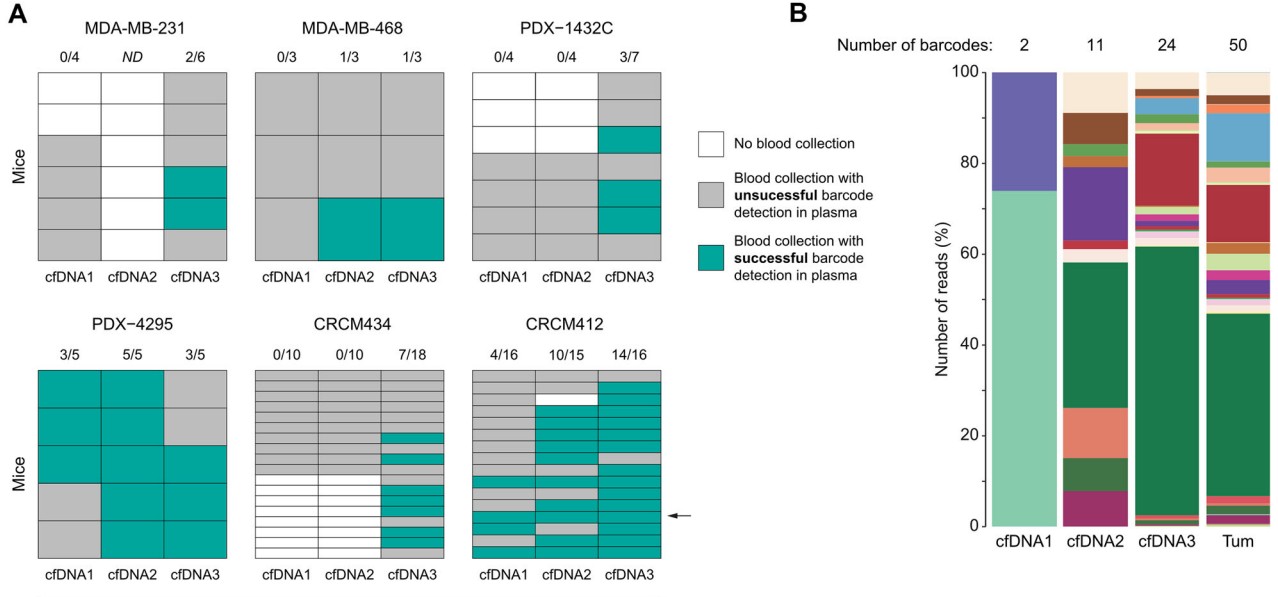

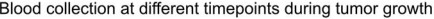

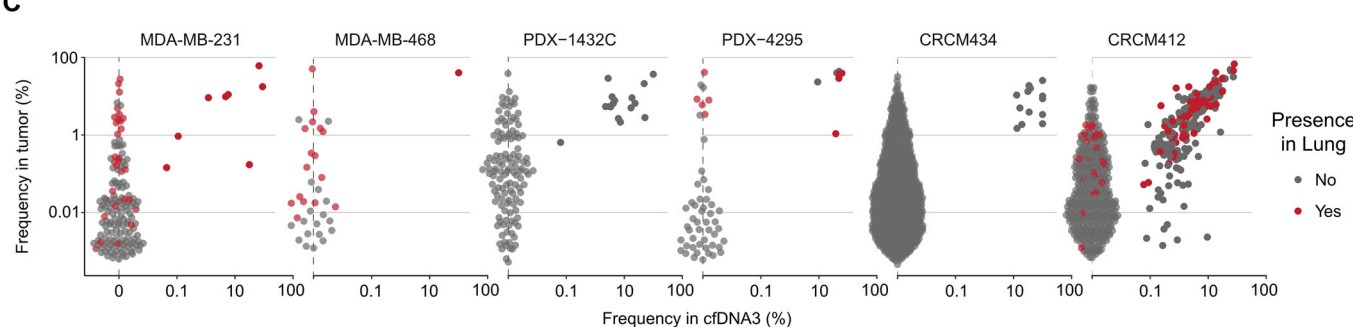

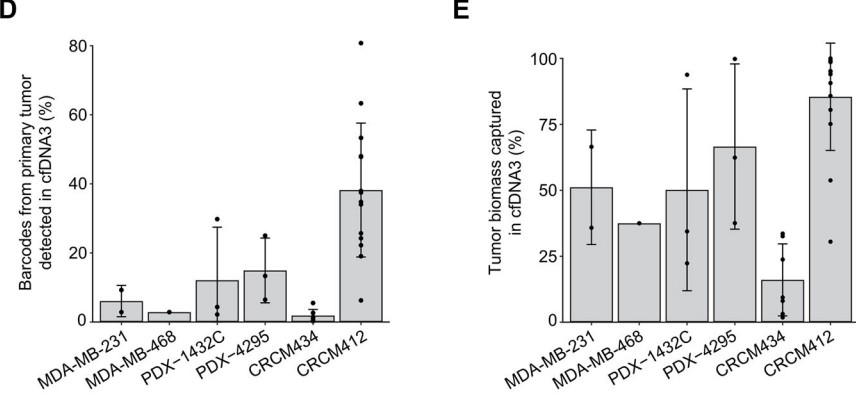

**Figure 3. Detection of barcodes in the plasma.**

(A) Barcode detection in cfDNA from early bleeding (cfDNA1 and cfDNA2) or terminal end bleed (cfDNA3). Unsuccessful barcode detection is highlighted in gray, and successful barcode detection in green. The numbers of successful recoveries at each time point are indicated on the top. (B) Frequency of barcodes (based on number of reads) detected in cfDNA samples indicated with an arrow in panel (A) and its associated primary tumor. Each color represents a barcode. (C) Barcode frequency in primary tumors and cfDNA. Each dot represents a barcode. The barcodes represented in red were also detected in the lungs. (D) Percentage of barcodes present in the primary tumor and detected in cfDNA3. (E) Percentage of primary tumor biomass captured in cfDNA3. For panels (C–E), only samples in which barcodes were detected in the plasma were included. (D, E) Each dot corresponds to an independent tumor. The error bars represent the standard deviation (SD) of the mean. MDA-MB-231, one experiment, $n = 2$, MDA-MB-468 one experiment, $n = 1$, PDX-1432C two independent experiments, $n = 1, 2$ PDX-4295 one experiment, $n = 3$, CRCM434 three independent experiments, $n = 3, 2, 2$, CRCM412, four independent experiments, $n = 4, 2, 4, 4$. Source data are available online for this figure.

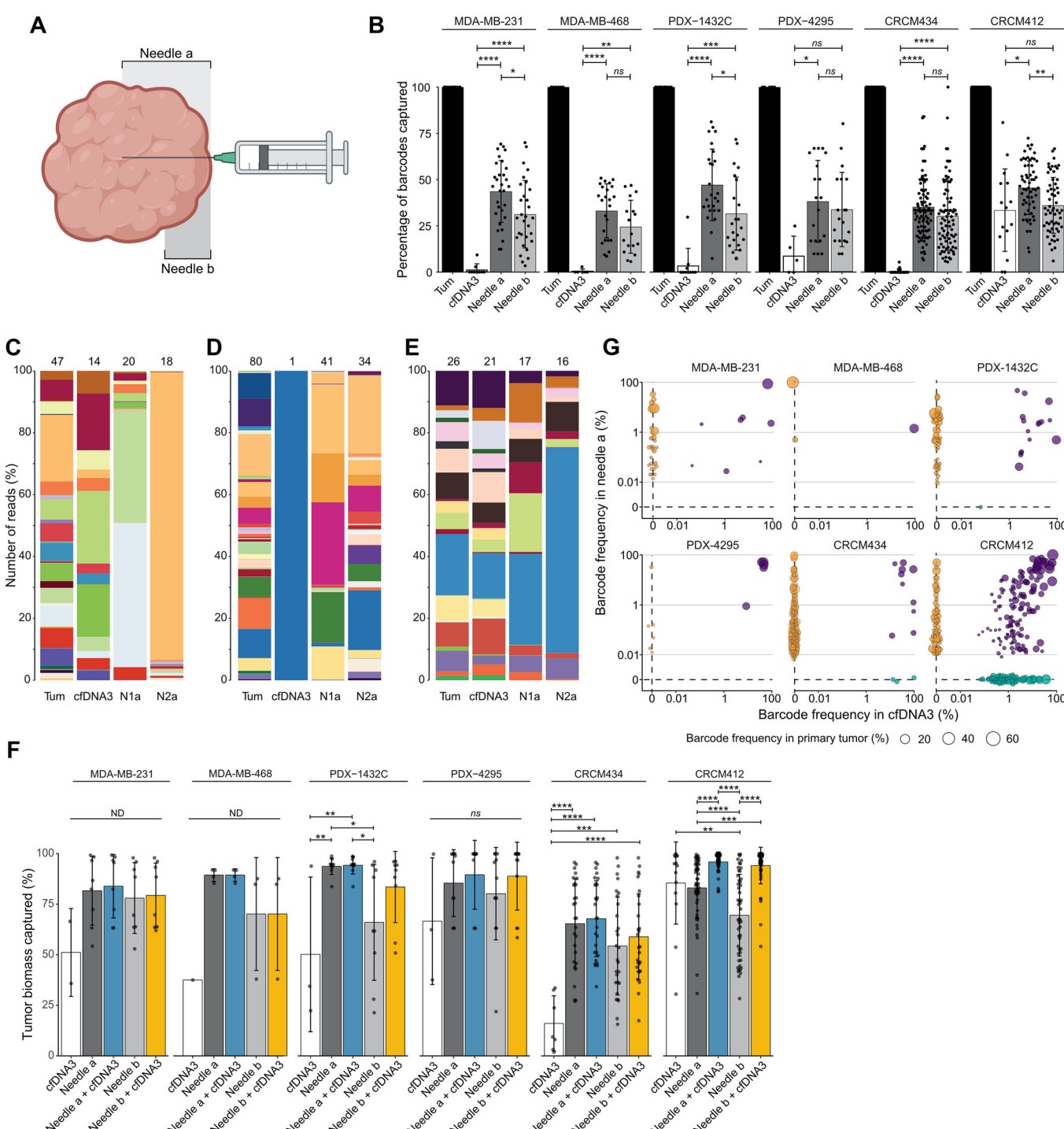

in plasma), except for CRCM412, where cfDNA overall captured more tumor biomass than the two needle samplings.

We then assessed whether the same clones were captured by both methods and whether combining several biopsies conferred a quantitative advantage in assessing tumor clonality. Barcodes captured by both cfDNA and solid biopsies were often similar (Fig. 4G, purple dots). In comparison, solid biopsies seemed to have more unique barcodes (Fig. 4G, yellow dots on the *y* axis) and were more efficient at capturing minor barcodes in all models except CRCM412 (Fig. EV6C).

When two needle samples were combined from the same primary tumor, the overall primary tumor biomass captured slightly increased (Fig. EV6D). Furthermore, combining cfDNA and needle biopsies significantly increased the percentage of biomass captured compared to cfDNA alone (Figs. 4F and EV6E), with a few exceptions. In addition, the combination of solid and liquid biopsies significantly increased the percentage of biomass captured in CRCM412 (Fig. 4F). However, it didn't provide any significant advantage in the other models (Figs. 4F, EV6E, and EV7A).

**Figure 4. Comparison of the barcode repertoire captured in liquid and solid biopsies.**

(A) Schematic overview of needle depth during sampling in a primary tumor. (B) Percentage of barcodes detected in needle biopsies from the primary tumor (plotted as reference at 100%) and cfDNA. In this panel, all the cfDNA samples were included, including those with unsuccessful barcode recovery. Needle-a corresponds to solid biopsy reaching the center of the tumor (deep needle sampling). Needle-b corresponds to shallow sampling, covering only a quarter of the tumor diameter. Each dot corresponds to a biopsy sample. Significance from one-way ANOVA followed by Tukey multiple comparisons test. $P$ values: MDA-MB-231 cfDNA3/Needle-a $= 1.3e-08$, cfDNA3/Needle-b $= 2.8e-05$, Needle-a/Needle-b $= 0.0139$; MDA-MB-468 cfDNA3/Needle-a $= 1.2e-05$, cfDNA3/Needle-b $= 0.0017$; PDX-1432C cfDNA3/Needle-a $= 1.0e-07$, cfDNA3/Needle-b $= 0.0005$, Needle-a/Needle-b $= 0.0130$; PDX-4295 cfDNA3/Needle-a $= 0.0187$; CRCM434 cfDNA3/Needle-a $= 4.1e-14$, cfDNA3/Needle-b $= 1.5e-11$; CRCM412 cfDNA3/Needle-a $= 0.0195$, Needle-a/Needle-b $= 0.0037$. (C–E) Three representative examples of the clonal repertoire detected in primary tumors, cfDNA3, and two needles sampling in PDX models: (C) PDX-1432C, (D) CRCM434, and (E) CRCM412. In these stacked histograms, each barcode is represented by a color, and the number of barcodes per sample is indicated at the top of the histogram. (F) Percentage of the primary tumor biomass detected in different biopsies: cfDNA (white), Needles a (dark gray), combination of deep needle sampling plus cfDNA3 (blue), Needles b (light gray), combination of shallow needle sampling plus cfDNA (yellow). Each dot corresponds to a biopsy sample. Significance from one-way ANOVA followed by Tukey multiple comparisons test. $P$ values: PDX-1432C cfDNA3/Needle-a $= 0.0084$, cfDNA3/Needle-a+cfDNA3 $= 0.0075$, Needle-a/Needle-b $= 0.0192$, Needle-a+cfDNA3/Needle-b $= 0.0165$; PDX-4295 cfDNA3/Needle-a $= 0.0187$; CRCM434 cfDNA3/Needle-a $= 2.7e-06$, cfDNA3/Needle-a+cfDNA3 $= 7.7e-07$, cfDNA3/Needle-b $= 0.0004$, cfDNA3/Needle-b+cfDNA3 $= 5.8e-05$; CRCM412 cfDNA3/Needle-b $= 0.0020$, Needle-a/Needle-a+cfDNA3 $= 5.1e-05$, Needle-a/Needle-b $= 2.0e-05$, Needle-a/Needle-b+cfDNA3 $= 0.0008$, Needle-b/Needle-b+cfDNA3 $= 5.8e-14$, Needle-b/Needle-a+cfDNA3 $= 5.7e-14$. (G) Relationship between barcodes in solid biopsy (Needle 1) and cfDNA (cfDNA3), depending on the barcode frequency in primary tumor (represented by the size of the dots). Each dot represents a barcode. Barcodes detected within the two biopsies are highlighted in purple, and barcodes exclusively detected by a method are plotted at 0, needle (orange), and cfDNA (green). For panels (C–G), only samples in which barcodes were detected in the plasma were included. (B, F) Significance from one-way ANOVA followed by Tukey multiple comparisons test, ns non-significant, $P$ value $> 0.05$, *$P$ value $< 0.05$, **$P$ value $< 0.01$, ***$P$ value $< 0.001$, ****$P$ value $< 0.0001$. Error bars represent the standard deviation (SD) of the mean. (B) MDA-MB-231, three independent experiments, $n = 1, 4, 4$, MDA-MB-468 one experiment, $n = 6$, PDX-1432C two independent experiments, $n = 6, 4$, PDX-4295 two experiment, $n = 1, 4$, CRCM434, four independent experiments, $n = 7, 6, 4, 5$, CRCM412, four independent experiments, $n = 4, 2, 5, 5$. (F, G) MDA-MB-231 one experiment, $n = 2$, MDA-MB-468 one experiment, $n = 1$, PDX-1432C two independent experiments, $n = 1, 2$, PDX-4295 one experiment, $n = 3$, CRCM434 three independent experiments, $n = 3, 2, 2$, CRCM412 four independent experiments, $n = 4, 2, 4, 4$. Source data are available online for this figure.

Lastly, in the MDA-MB-231 model, CTCs were successfully retrieved from blood, and the barcode repertoire was analyzed (Fig. EV7B). While barcodes detected when combining needle biopsies and cfDNA already accounted for a high percentage of the primary tumor biomass, adding barcodes collected from CTCs further improved the assessment of ITH, covering over 80% of tumor biomass (Fig. EV7B).

# Discussion

In clinical settings, probing ITH is of utmost importance for the ongoing development of precision medicine. As drug resistance and metastasis can be driven by minor clones present in primary or secondary lesions (Hu et al, 2020a), there is a need to capture molecular information from a large number of clones, regardless of their frequency, to guide the choice of optimal therapies. Here, genetic barcoding was used to generate clonally structured tumors in multiple mice, offering a robust qualitative and quantitative assessment of the extent of heterogeneity captured in solid and liquid biopsies using two breast cancer cell lines and four PDXs. Overall, the superiority of one sampling method over another clearly depends on the model and tumor burden. This suggests that, in the clinic, combining tumor and blood sampling could provide a better assessment of ITH.

Barcoded clones were likely subject to genetic drift over time and passaging, as previously demonstrated with cell lines and PDXs, both in vitro (Ben-David et al, 2018) and in vivo (Eirew et al, 2015). By focusing on barcode analysis rather than genomic heterogeneity, this study may underestimate the extent of diversity captured in primary tumors and biopsies. It may also highlight a level of heterogeneity that is not clinically relevant to guide therapeutic decisions. However, the main advantage of this strategy is that it offers the capacity to track labeled cells and their progeny across primary tumors and subsequent sampling in an unbiased,

controlled, and comprehensive manner, using multiple mice per cohort. In this context, future studies integrating barcode detection and genomic analysis will provide additional insights into clonal evolution and its impact on the extent of genomic heterogeneity captured in biopsies. These models could also be used to improve methods to detect tumor material in liquid biopsies (based on cfDNA, methylation, and CTCs) and to assess the impact of specific therapies on DNA shedding.

Our results suggest that solid biopsies can capture a wide range of minor and dominant clones present in primary tumors across models, representing ~60–90% of the primary tumor biomass. However, the clonal repertoire captured in solid biopsies is strongly biased by the spatial distribution of clones within primary tumors. As a result, multiple sampling of a given tumor is likely to yield highly variable results in heterogeneous tumors. While it would be interesting to determine whether this cellular heterogeneity (based on barcode identification) correlates with genomic heterogeneity, this result corroborates previous observations from genomic analysis of serial biopsies (Pereira et al, 2021). Regional sequencing of patient tumors (de Bruin et al, 2014; Gerlinger et al, 2012; Navin et al, 2011; Shah et al, 2012; Yates et al, 2015) and recent special transcriptomic analysis (Mo et al, 2024; Moehlin et al, 2021) also support the observation that tumor sampling is not necessarily representative of the whole tumor heterogeneity.

In the clinic, needles are often directed toward the center of a tumor. Indeed, both computational modeling and empirical data from cellular barcoding experiments highlighted that the center of non-necrotic tumors is significantly enriched in barcodes compared to the periphery. While a similar observation has been made in various cancer types, using genomic analysis and modeling (Chkhaidze et al, 2019; Lewinsohn et al, 2023; Noble et al, 2022; van der Heijden et al, 2019), it would be important to confirm this in immunocompetent preclinical models, in models that are not relying on tumor transplantation, and in patient samples. This observation is extremely timely, as many studies are currently

investigating the transcriptomic profile of cancer clones using spatial transcriptomics (de Vries et al, 2020; Mo et al, 2024; Moehlin et al, 2021; Wu et al, 2021). It will be important to determine whether this difference in clonal frequency and distribution can be attributed to particular features of the clones. Furthermore, cells from each subclone may have distinct gene expression profiles depending on their localization within the tumor. Breast cancer stem cells from the center, for instance, were shown to be more epithelial than cancer stem cells present in the invasive front (Liu et al, 2014). Further analysis using spatial transcriptomics, optical barcoding, and time-lapse imaging will allow the study of these mechanisms over time.

In parallel, a quantitative comparison between barcodes in mouse plasma and those in whole tumors suggested that, when detected, cfDNA can be a good surrogate for tumor heterogeneity, as it captured up to 80% of the tumor biomass. In these settings, the likelihood of detecting barcodes in cfDNA correlated with the extent of tumor burden, and dominant barcodes present in primary tumors and lungs had a higher likelihood of being detected in cfDNA, corroborating previous patient studies (Dawson et al, 2013; Murtaza et al, 2015; Namløs et al, 2018; Parikh et al, 2019). It is also possible that necrotic models (such as PDX-1432C) were more likely to shed DNA into the vasculature compared to non-necrotic models (such as MDA-MB-231 xenografts). Nonetheless, discrepancies were identified between models, and tumor burden and extent of necrosis were not solely responsible for these differences. In the case of PDX CRCM412, for instance, cfDNA can be detected at early time points, despite its low-necrotic grade. Interestingly, cells were detected in the lungs in this model, and it might be that tumors able to shed cancer cells into the bloodstream are also more likely to shed cfDNA. It would be interesting to validate this hypothesis and determine whether the presence of previously described 'shedders' and 'seeders' (Merino et al, 2019) correlates with the presence of cfDNA in the plasma. Understanding the mechanisms involved in cfDNA shedding will be clinically relevant.

cfDNA has a promising role in the quantitative monitoring of early recurrence and tumor progression, and it also has qualitative utility as a molecular profiling tool. Tests such as FoundationOne Liquid CDx have been FDA-approved to guide therapeutic interventions in several types of cancer (non-small cell lung cancer, breast, prostate, and ovarian cancers) based on a panel of somatic and germline mutations. Therefore, understanding the behavior and molecular profile of clones likely to be detected in plasma will be useful for interpreting liquid biopsies. While our results indicate that barcodes from cfDNA predominantly represent dominant clones from the primary tumors, it would be interesting to determine whether these clones have seeding properties (Merino et al, 2019) and might, therefore, be responsible for the establishment of metastases. To study the ability of cfDNA to reflect the heterogeneity of metastases, the barcode repertoire detected in plasma could be compared to metastases after resection of the primary tumor. In this case, cfDNA may be a good surrogate of the biomass of metastases from different sites that are difficult to sample with solid biopsies.

The qualitative analysis of barcodes captured in the plasma of multiple models suggested that dominant clones were represented, but clones that were under-represented in the primary tumor mright not be detected. This could be due to several technical limitations. First, experiments with xenograft models are often short in duration. This timeframe doesn't allow for follow-up comparable to clinical settings, regarding clonal evolution and dynamics. Second, these results, which are based on the detection of genetic barcodes smaller than 100 bp, may underestimate the representation of cfDNA fragments detected in clinical settings (Dawson et al, 2013). It is possible that some barcodes present in truncated forms were not recognized during primer annealing and therefore not detected. In clinical settings, detecting multiple gene fragments in the plasma is likely to provide a better representation of tumor heterogeneity, a superior detection of minor clones, and an improved sensitivity in cases of low tumor burden. Finally, the volumes of blood and the bleeding strategies used in these preclinical models differed from those in clinical settings. However, while this quantitative analysis in xenograft models, using barcode detection, might not be directly comparable to analysis of patient biopsies, it provides an opportunity to study the process of DNA shedding by cancer clones longitudinally, in the absence of therapeutic interventions. Interestingly, variability in liquid biopsies was observed in mice bearing tumors with similar clonal composition, suggesting that some of these variations may be stochastic. Additional studies linking genomic analysis and clonal information in these models will be required to better understand the process of DNA shedding and why some models, such as PDX CRCM412, shed a significant amount of cfDNA, while others, such as MDA-MB-231, do not, despite being highly metastatic. Such investigations will be required to optimize cfDNA use in disease monitoring, for instance, via the identification of biomarkers associated with false negatives, by characterizing aggressive clones that do not shed cfDNA in the plasma.

Finally, our results demonstrated that combining liquid and solid biopsies can provide a significant advantage in assessing ITH, depending on the tumor model. This observation supports other studies indicating that it might be difficult to substitute one type of biopsy with the other (Esagian et al, 2020; Parikh et al, 2019), and that ensuring that both techniques coexist would improve breast cancer diagnosis and management (Finzel et al, 2018; Pesapane et al, 2020). Indeed, sequencing analysis of 351 samples from patients with diverse cancer types suggested that the combination of both solid and liquid biopsies offers a more therapeutically valuable representation of tumor heterogeneity in clinical settings (Finzel et al, 2018). Similar studies comparing the utility of both liquid and solid biopsies in clinical settings will be required to optimize their use, not only in the context of tumor heterogeneity, but also to identify the risks of disease progression and drug resistance. Furthermore, solid biopsies present the advantage of capturing intact cells, increasing the scope of downstream applications in diagnostics and research from cellular assays to studying malignant or normal cells (Bianchini et al, 2010; Deng et al, 2020), to multi-omics analysis based on bulk or single cells (Kim et al, 2018; Li et al, 2021). From a diagnostic and prognostic perspective, they provide a unique platform to analyze clinically relevant markers that are complementary to the study of cfDNA, such as tumor-infiltrating lymphocytes, which are predictive of immunotherapy efficacy. In this context, the analysis of single cells in blood might provide additional information.

While more work will be required to better understand the properties and dynamics of cancer clones across different types of biopsies over time and in response to specific therapies, this study provides new insights into the utility of barcoded models for

studying the variability and biases of solid and liquid biopsies. Linking meaningful clonal information to multi-omics analyses in preclinical models of cancer holds great promise in the development of diagnostic tools and the implementation of personalized medecine.

# Methods

### Reagents and tools table

| Reagent/resource | Reference or source | Identifier or catalog number |
| --- | --- | --- |
| **Experimental models** | | |
| MDA-MB-231 | ATCC | HTB-26 |
| MDA-MB-468 | ATCC | HTB-132 |
| NSG (*M. musculus*) | Jackson Lab | NOD.Cg-Prkdcscid Il2rgtm1Wjl/SzJ (Strain #:005557) |
| **Cell culture reagents** | | |
| DMEM/F12 | ThermoFisher | 10565042 |
| RPMI 1640 | ThermoFisher | 22400086 |
| DPBS | ThermoFisher | 14040133 |
| TrypLE Express | ThermoFisher | 12604013 |
| **Oligonucleotides and other sequence-based reagents** | | |
| PCR primers | This study | Dataset EV2 |
| **Chemicals, enzymes, and other reagents** | | |
| Collagenase IA | Sigma-Aldrich | C9891 |
| Hyaluronidase | Sigma-Aldrich | H3506 |
| DNAse I | Worthington | LS002139 |
| B27 | ThermoFisher | 17504001 |
| Penicillin–streptomycin | ThermoFisher | 15140122 |
| Insulin | Sigma-Aldrich | 11376497001 |
| Hydrocortisone | Sigma-Aldrich | H0396-100MG |
| Heparin | Sigma-Aldrich | H0878 |
| Fibroblast growth factor | Merck-Millipore | 01-106 |
| Epidermal growth factor | Sigma-Aldrich | E9644 |
| DirectPCR Lysis reagent | ViagenBiotech | 303-C |
| Proteinase K | ThermoFisher | 25530049 |
| DNA Polymerase | NEB | M0273E |
| Standard Taq Buffer | NEB | B9015S |
| MgCl2 | NEB | B9021S |
| dNTPs | NEB | N0447L |
| UltraPure Distilled Water | ThermoFisher | *10977-015* |
| QIAamp Circulating Nucleic Acid Kit | Qiagen | 55114 |
| Propidium iodide | ThermoFisher | P1304MP |
| Magnetic beads | Macherey-Nagel | 744100.4 |
| 4% PFA | ThermoFisher | 28908 |
| Agarose | Bioline | BIO-41025 |

| Reagent/resource | Reference or source | Identifier or catalog number |
| --- | --- | --- |
| Gel loading dye | NEB | B7025S |
| DNA Ladder | NEB | N0557S |
| **Software** | | |
| Affinity designer 2 | Serif (Europe) Ltd | |
| R 4.5.1 (2025-06-13 ucrt) | Comprehensive R Archive Network (CRAN) | |
| RStudio 2025.09.1 + 401 | Posit Software, PBC | |
| **Other** | | |
| 24-well plate flat-bottom ultralow attachment | Corning | 734-1584 |
| 10 cm cell culture dishes | Corning | 353003 |
| Needles 23 G | Terumo | TE8-2325 |
| Insulin Syringe 1 ml 27 G | Terumo | TE-10M2713 |
| Microvette tubes | Sarstedt | *20.1341.100* |
| 70 uM Strainer | Greiner | 542070 |
| 96-well plates | SSIbio | 3420-00S |
| **Instruments** | | |
| Flow cytometry | BD | FACSAria III |
| Thermal Cycler | Bio-Rad | abS100 |
| Sequencer | Illumina | NextSeq500 |

### In silico modeling of 3D growth

Simulations of 3D growth of barcoded tumor cells were performed as previously described (Merino et al, 2019). In brief, the simulation code by Waclaw B et al (Waclaw et al, 2015) was adapted to account for cellular barcoding and transplantation into the mammary fat pad, using the parameters originally proposed by the authors for growth and migration. Simulations were initiated with 200 barcoded tumor-initiating cells and ran until reaching a size of 10 million cells. To quantify clonal density, the simulated tumor was split in silico into five pieces (as shown in Fig. 1B), and the number of clones counted and their frequencies were plotted. The positions of each cell as well as clonal identity (color) were rendered in 3D using the open source visualization tool Ovito (Stukowski, 2010).

### PDX establishment and amplification

PDX-1432C was established at the ONJCRI from a triple-negative treatment naïve breast cancer tumor. KCC-P-4295, referred to in the manuscript as PDX-4295, was obtained from BROCADE and established at the Kinghorn Cancer Centre and Garvan Institute from a treatment naïve TNBC. PDXs CRCM434 and CRCM412 were generated at the Institut Paoli-Calmettes from drug naïve TNBC patient tumors. All PDXs were orthotopically injected into the mammary fat pad of females NOD-SCID-IL2Rγ−/− (NSG) to be amplified prior to barcoding. All procedures in animals were conducted in accordance with the National Health and Medical

Research Council guidelines under the approval of the Austin Animal Ethics Committee. The use of patient samples was approved by the Austin Health Human Research Ethics Committee.

PDXs tumors were harvested and prepared as a single-cell suspension. The tissues were manually chopped into small pieces (about 1 mm by 1 mm) and resuspended for 1 h in the following digestion medium: collagenase IA (300 U/ml) (#C9891, Sigma-Aldrich), hyaluronidase (100 U/ml) (#H3506, Sigma-Aldrich), and deoxyribonuclease I (DNase I) (100 U/ml) (#LS002139, Worthington) in DMEM/F12 (#10565042, ThermoFisher). PDXs cells were plated in 24-well plates (flat-bottom ultralow attachment, #734-1584, Corning) at a density of 300,000 cells in 300 μl of mammosphere media. Mammosphere medium was composed of DMEM-F12 (#10565042, ThermoFisher) supplemented with 1× B27 (#17504001 ThermoFisher), 100 U/ml of penicillin–streptomycin (#15140122, ThermoFisher), 5 μg/ml insulin (#11376497001, Sigma-Aldrich), 1 ug/ml hydrocortisone (#H0396-100MG, Sigma-Aldrich), 0.8 U/ml heparin (#H0878-100 KcU, Sigma-Aldrich), 20 ng/ml basic fibroblast growth factor (#01-106, Merck-Millipore), and 20 ng/ml epidermal growth factor (#E9644, Sigma-Aldrich).

## Genetic barcoding experiments

For PDXs, cells were infected with lentiviruses containing the barcode library as previously described (Merino et al, 2019), at low MOI (PDX-1432CC 8.7 ± 1.1%, CRCM434 6.4 ± 2.9%, PDX-4295 7.5 ± 3.5%, CRCM412 5 ± 1.5% (mean ± SD)) to ensure the integration of a single barcode per cell. Barcoded cells were sorted for GFP positivity and resuspended in injection buffer (42.5% DPBS, 30% FBS 25% Matrigel, and 2.5% Trypan blue). In total, 2.5–5k of cells were injected into the fourth mammary gland of NSG mice. For PDX CRCM412, the barcode composition of primary tumors and metastases from mice 88–92 have been previously described (Serrano et al, 2023). To generate the models used for clone-splitting experiments, barcoded tumors from PDX-1432C were harvested once they reached 300 mm$^3$, processed into a single-cell suspension, and reinjected in 12 recipient mice.

For the cell lines, cells were obtained from the ATCC, MDA-MB-231 (#HTB-26, passage 39), and MBA-MB-468 (#HTB-132, passage 33) were maintained in RPMI 1640 with HEPES (#22400086, ThermoFisher), 10% fetal bovine serum (FBS), and penicillin–streptomycin (#15140122, ThermoFisher) at 10,000 U/ml. For barcode infection, 2.2 million cells were plated in 10-cm cell culture dishes (#353003, Corning) with 8 ml culture media. Lentiviruses were added for 48 h h. Barcoded cells were selected via flow cytometry based on GPF positivity, both populations under 0.1 MOI, 7.1% and 5% for MDA-MB-231 and MDA-MB-468, respectively. In total, 25,000 cells per cell line were then expanded in vitro. 200k MDA-MB-231 barcoded cells and 300k MDA-MB-468 cells were injected into the mammary fat pad of NSG mice. Mice 36–39, 69–72, 88–92 were previously analyzed for barcode composition in primary tumors and metastases (Serrano et al, 2023).

## Tumor processing and needle biopsies

Genetically barcoded tumors were harvested once they reached 800 mm$^3$, prior to needle sampling. A preloaded 3-ml syringe with 500 μl DPBS fitted with a 23 G needle was inserted in the primary tumor with minimal aspiration to ensure the capture of tissue. The biopsy content was collected in a 1.5-ml Eppendorf tube. Needles and syringes were changed between each biopsy. Tubes containing

biopsy samples were spun down 5 min at 500 rpm, and the supernatant was removed. The pellets were resuspended in 50 ul lysis buffer (Viagen Biotech) with 1:50 proteinase K 20 mg/mL (Invitrogen) and lysed 1 h at 55 °C followed by 30 min at 85 °C, and finally 5 min at 95 C on a heater shaker dry bath set to 800 rpm. Samples were stored in a freezer until PCR amplification.

Tumor dissections were performed with surgical blade n°10 in order to isolate tumor center from edges as shown in Fig. 1C. Edges on the transversal axes were cut first from the primary tumor, then the lateral edges were removed from the primary tumor. The tumor was then flipped horizontally to dissect the superior and inferior edges, leaving the center of the primary tumor exposed. Pieces were re-cut if necessary to obtain pieces of equal size. Blades were changed between each tumor to avoid barcode cross-contamination. Pieces were resuspended in 300 μl of lysis buffer and lysed overnight on a heater shaker set to 800 rpm at 55 °C, then 85 °C 30 min and 95 °C for 5 min. Samples were stored in a freezer until PCR amplification.

## Blood collection and cfDNA isolation

Blood from mice was collected to isolate cfDNA. Terminal end bleed was performed via cardiac puncture after death with a 1 ml 27 G insulin syringe, 800 μl to 1 ml of blood was collected in microvette tubes (Sarstedt). For early time points, 200 μl of blood was collected via the lateral tail vein with a 0.5 ml 27 G insulin syringe and transferred to a heparin-coated tube. Blood samples were immediately processed. Blood tubes were spun down at 1600× g for 10 min in a "swing-out" rotor centrifuge set with the brakes off. Plasma was transferred to 1.5-ml tubes and centrifuged for 5 min at 14,000 rpm to pellet cellular debris. The supernatant was transferred into a new tube and stored at −80 °C until cfDNA extraction. cfDNA extraction was performed using the QIAamp Circulating Nucleic Acid kit (Qiagen).

## Lung metastasis analysis

Lungs were collected at the ethical endpoint. The tissue was manually chopped, and the digestion was done in 5 ml of RPMI 1640 for MDA-MB-231 and MDA-MB-468 cell lines, and DMEM F12 (Thermo-Fisher) for PDX cells. Both media were supplemented with 300 U/ml collagenase IA (Sigma-Aldrich), 100 U/ml hyaluronidase (Sigma-Aldrich). Samples were incubated at 37 °C on an orbital shaker set at 300 rpm for 45 min, then resuspended through a 18-G needle after 20 min, and a 21-G needle after 40 min of digestion. The cell suspension was filtered through a 70 μm cell strainer and spun down for 5 min at 500× g. PDX lung samples were resuspended in DPBS with PI to be sorted via flow cytometry. Sorted cancer cells from the whole lungs were spun down and resuspended in 50 μl lysis buffer. Cells from the cell lines were resuspended in 100 μl lysis buffer. Samples were lysed for 1 h at 55 °C followed by 30 min at 85 °C, and finally for 5 min at 95 °C on a heater shaker set to 800 rpm. Samples were stored in a freezer until PCR amplification.

## Barcode amplification and sequencing

PCR amplification was performed on crude lysates; tumor pieces were diluted 1:10 in water. In total, 40 μl of this template were mixed with 160 μl of PCR mix in a 96-well plate (#3420-00S,

SSIbio) and split into two replicates of 100 µl before the start of the PCR to assess barcode detection reliability. cfDNA and lung samples were run in quintuplicate. Primers are included in Dataset EV2. The first PCR included common primers (TopLib 5'-TGCTGCCGTCAACTAGAACA-3' and BotLib 5'-GATCTC-GAATCAGGCGCTTA-3') to allow for barcode amplification. Cycle specification was at 94 °C for 5 min, followed by 30 cycles at 94 °C for 15 s, 57.2 °C for 15 s, 72 °C for 15 s, and then 72 °C for 10 min. The product of the first PCR was then used to run the second PCR, to add specific individual indexes for NexGen sequencing (Dataset EV2). Cycle specifications of the thermocycler were 94 °C for 5 min, followed by 30 cycles at 94 °C for 5 s, 57.2 °C for 5 s, 72 °C for 5 s, and then 72 °C for 10 min. All PCRs were run on abS100 thermal cycler (Bio-Rad), and the final presence of PCR product at 266 bp was verified on 2.5% agarose gel electrophoresis. Samples were pooled and clean-up with magnetic beads (#744100.4, Macherey-Nagel) before sequencing on Next-Seq (Illumina).

## Histology

Tissues were fixed in 4% paraformaldehyde for 48 h before transfer to 70% Ethanol, for block embedding, and staining was performed by the Department of Pathology at the Austin Health Hospital. Slides were scanned on Aperio AT2.

## Bioinformatic analysis

Sequencing results were analyzed on RStudio (Version 1.4.1106), and demultiplexing of sequencing FastQ files was performed using the ProcessAmplicon function from the edgeR package (https://doi.org/10.18129/B9.bioc.edgeR) to generate a read-count matrix for each barcode per sample. To ensure the quality of the data, the dataset was filtered as follows. First barcode read counts less than or equal to 10 were set to zero within each sample. Next, samples with fewer than 10000 total reads were excluded from further analysis. Then, replicate and quintuplicate samples with a Pearson correlation inferior to 0.6 were removed from the analysis (except for cfDNA samples). Finally, barcodes present in less than two replicates were discarded. Replicates were then pooled by adding the read-count values of the barcode and normalized. The value of each barcode within individual tumor pieces were cumulatively added and normalized to recreate the profile of the full tumor as displayed in Fig. 2C. Finally, barcodes only present in primary tumors were conserved across samples. All subsequent visualization and statistical analysis were performed in R. When $n < 3$, statistical analyses were not determined (ND).

## Data availability

Barcode sequencing data have been deposited in the Sequence Read Archive (SRA) under the accession number PRJNA1368999. Code to reproduce all relevant figures and supplementary figures included in this article is available at https://github.com/Anto-Ser/Biopsies_Barcoding.git.

The source data of this paper are collected in the following database record: biostudies:S-SCDT-10_1038-S44320-026-00194-w.

# Peer review information

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

## Acknowledgements

We would like to thank Robin Anderson, Bhupinder Pal, Caroline Bell, Sarah Ellis (ACRF Centre for Imaging of the Tumour Environment), David Baloyan (Flow Cytometry Core Facility), and Stephen Wilcox (WEHI Advanced Genomics Platforms) for their advice and technical assistance. The authors and Olivia Newton-John Cancer Research Institute gratefully acknowledge the generous support of the Love Your Sister Foundation. We are grateful to the patients who consented for their tissue to be donated, and for the support of the BROCADE Rapid Autopsy Program, funded by the Australian National Breast Cancer Foundation (NBCF) under infrastructure grant IF-14-001, to facilitate access to PDX-4295. We also acknowledge Lisa Devereux, Robin Anderson and Alex Swarbrick, who manage and lead BROCADE. We acknowledge the Kinghorn Cancer Center and Garvan Institute for the KCC-P-4295 model (PDX-4295), the Cancer Research Center of Marseille (CRCM) for PDXs CRCM412 and CRCM434, and Ton Schumacher and the Netherlands Cancer Institute for providing the genetic barcoding library. Part of the icons and artworks present in the synopsis and figures were created in BioRender. Serrano A (2025) https://BioRender.com/721mrg1. The Olivia Newton-John Cancer Research Institute acknowledges the support of the Operational Infrastructure Program of the Victorian Government. AS was supported by the Melbourne Research Scholarship, DM is supported by the Victorian Cancer Agency (MCRF21011), the NBCF (Investigator Initiated Research Grant IIRS0049), and NHMRC (GNT2012196 and 2027459). DM and BY are supported by the Love Your Sister Foundation. The authors acknowledge the ACRF Centre for Imaging the Tumor Environment at the Olivia Newton-John Cancer Research Institute for providing microscopy support, and Austin Pathology.

## Author contributions

**Antonin Serrano**: Conceptualization; Data curation; Software; Formal analysis; Validation; Investigation; Visualization; Methodology; Writing—original draft; Writing—review and editing. **Tom S Weber**: Conceptualization; Resources; Data curation; Software; Formal analysis; Supervision; Funding acquisition; Investigation; Visualization; Writing—original draft; Writing—review and editing. **Jean Berthelet**: Investigation; Writing—review and editing. **Sarah Ftouni**: Investigation; Writing—review and editing. **Farrah El-Saafin**: Investigation; Writing—review and editing. **Samuel Lee**: Data curation; Formal analysis; Writing—review and editing. **Elgene Lim**: Resources; Writing—review and editing. **Emmanuelle Charaffe-Jauffret**: Resources; Writing—review and editing. **Christophe Ginestier**: Resources; Writing—review and editing. **David Williams**: Investigation; Writing—review and editing. **Frédéric Hollande**: Supervision; Writing—review and editing. **Belinda Yeo**: Resources; Supervision; Funding acquisition; Writing—review and editing. **Sarah-Jane Dawson**: Conceptualization; Resources; Supervision; Funding acquisition; Writing—review and editing. **Shalin H Naik**: Conceptualization; Supervision; Funding acquisition; Investigation; Project administration; Writing—review and editing. **Delphine Merino**: Conceptualization; Resources; Supervision; Funding acquisition; Writing—original draft; Project administration; Writing—review and editing.

Source data underlying figure panels in this paper may have individual authorship assigned. Where available, figure panel/source data authorship is listed in the following database record: biostudies:S-SCDT-10_1038-S44320-026-00194-w.

## Disclosure and competing interests statement

The authors declare no competing interests.

# Expanded View Figures

**Figure EV1.   Analysis of the barcode repertoire in primary tumors.**

(**A**) Total number of barcodes detected in each tumor. Each dot represents a tumor sample. The average number of barcodes per tumor is indicated on top of the bars. (**B**) Shannon diversity index of barcoded tumors. Significance from one-way ANOVA followed by Tukey multiple comparisons test, *P* values: MDA-MB-231/PDX-1432C = 0.0070, MDA-MB-231/CRCM412 = 0.0320, MDA-MB-231/CRCM434 = 0.0003, MDA-MB-468/CRCM434 = 0.0443, PDX-1432C/PDX-4295 = 0.0254, PDX-4295/CRCM434 = 0.0022. *$P$ value < 0.05, **$P$ value < 0.01, ***$P$ value < 0.001, ****$P$ value < 0.0001. (**A**, **B**) Each dot corresponds to a tumor. Error bars represent standard deviation (SD) of the mean. MDA-MB-231 were shortened to MDA-231, and MDA-MB-468 were shortened to MDA-468. (**C–F**) Example of primary tumor cutting and barcode composition for one mouse per model, from top to bottom, MDA-MB-231, MDA-MB-468, CRCM434, PDX-4295, PDX-1432C, CRCM412. (**C**) Photos of tumors collected in PBS and cut into pieces of similar size. (**D**) Representation of the clonal composition of each piece. (**E**) Heatmap representing the clonal composition of each peripheral piece (normal font), center pieces (red) and full tumor (bold). Each column represents a barcode, and its frequency is represented by the color scale. (**F**) Relationship between the percentage of barcodes detected in each tumor piece and the weight of the tumor piece. Each dot corresponds to a tumor piece, from the periphery (purple dots) or the center (orange dots). (**A**, **B**) MDA-MB-231, three independent experiments, $n = 5, 4, 4$, MDA-MB-468 one experiment $n = 6$, PDX-1432C two independent experiments, $n = 6, 4$, PDX-4295 one experiment $n = 6$, CRCM434 four independent experiments, $n = 7, 6, 4, 5$, CRCM412 four independent experiments, $n = 4, 2, 5, 5$.

                                                                            

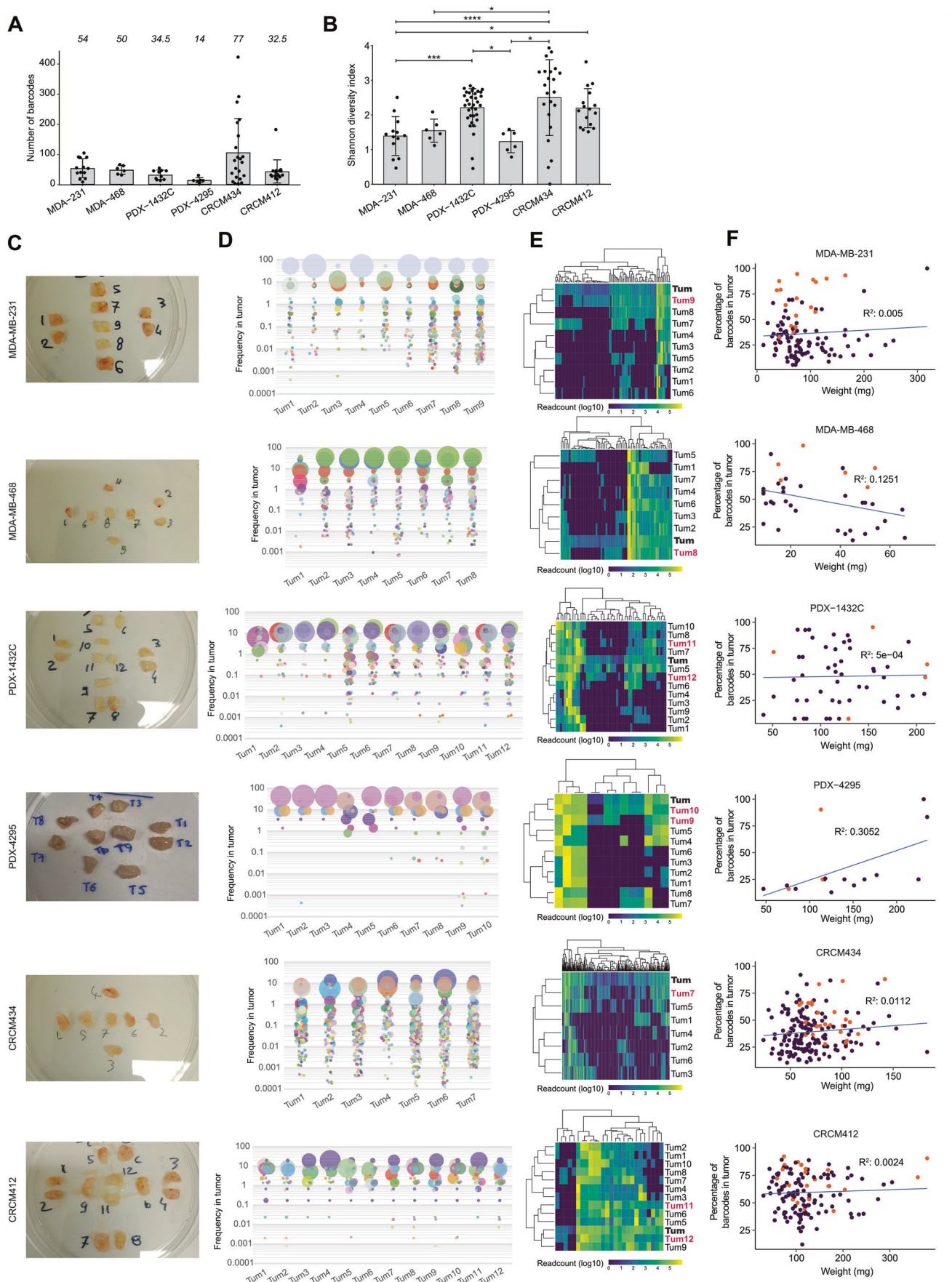

**A**

| ID | Sample | Subtype | Hormone status | | | Comments on necrosis |
|---|---|---|---|---|---|---|
| | | | ER | PR | HER2 | |
| 1432C | Patient | TNBC | Neg | Neg | Neg | Extensive necrosis |
| 1432C | PDX | TNBC | Neg | Neg | Neg | Extensively necrotic |
| CRCM412 | Patient | TNBC | Neg | Neg | Neg | *Not available* |
| CRCM412 | PDX | TNBC | Neg | Neg | Neg | Patchy necrosis |
| CRCM434 | Patient | TNBC | Neg | Neg | Neg | *Not available* |
| CRCM434 | PDX | TNBC | Neg | Neg | Neg | No comment on necrosis |
| 4295 | Patient | TNBC | Neg | Neg | Neg | *Not available* |
| 4295 | PDX | TNBC | Neg | Neg | Neg | No comment on necrosis |

**B**

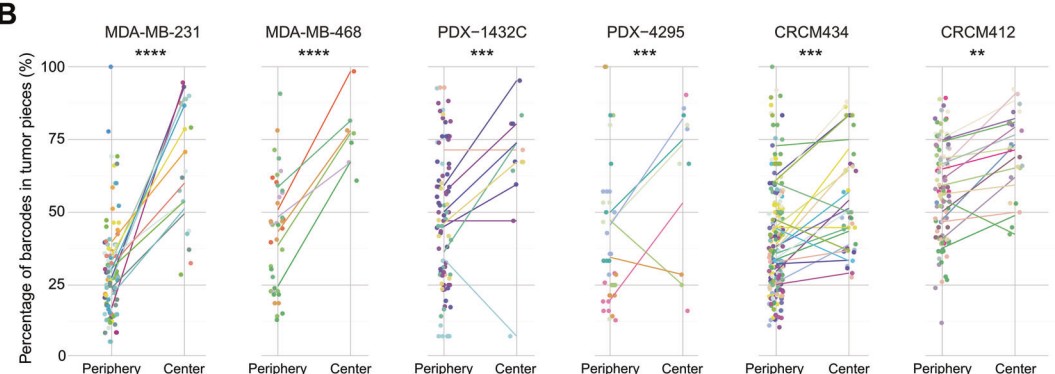

**C**

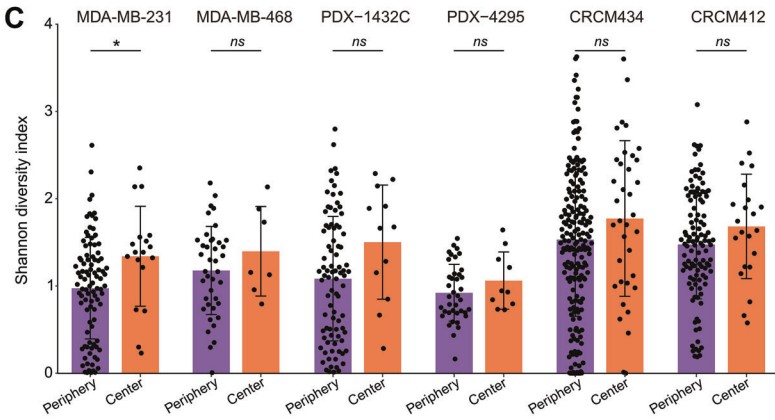

**D**

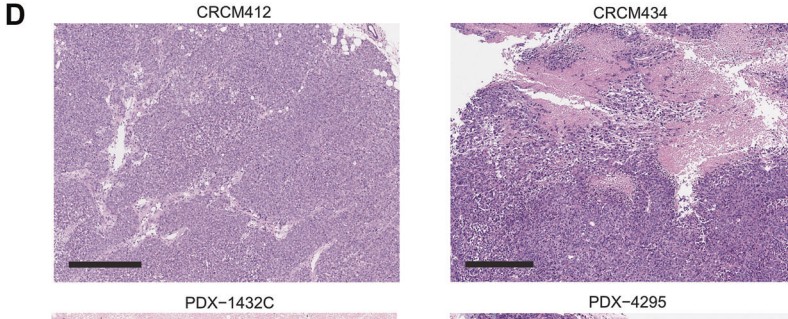

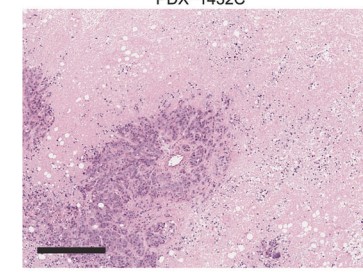

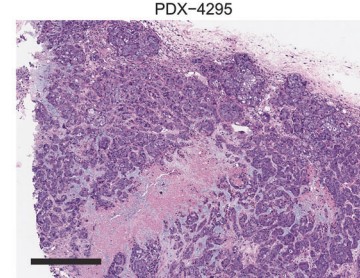

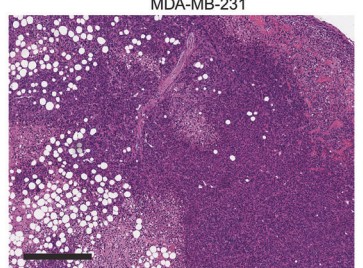

◀ **Figure EV2.   Characterisation of primary tumors.**

(**A**) Breast cancer subtype and hormone receptors status from patient samples and matching PDXs. Estrogen receptor (ER), Progesterone receptor (PR) Human epidermal growth factor receptor 2 (HER2). Comments on necrosis were shown when available on patient's pathology reports and PDXs. Non-barcoded primary tumors from previous PDX passages were used for analysis. (**B**) Mixed-effect model analysis of barcode distribution across tumor locations and models. Each color represents a specific mouse, each dot corresponds to a tumor piece, and the line represents the linear interpolation between average for each tumor. Pairwise Wald *t* tests (linear mixed-effects model) between "Center" and "Periphery" locations within each tumor model, *P* values: MDA-MB-231 = 2.7e-16, MDA-MD-468 = 7.7e-07, PDX-1432C = 0.0007, PDX-4295 = 0.0005, CRCM434 = 0.0002, CRCM412 = 0.0057. *$P$ value < 0.05, **$P$ value < 0.01, ***$P$ value < 0.001, ****$P$ value < 0.0001. (**C**) Shannon diversity index of tumor pieces from periphery (purple) or center (orange). Student's unpaired *t* test, *P* values: MDA-MB-231 = 0.0203, MDA-MD-468 = 0.3239, PDX-1432C = 0.0576, PDX-4295 = 0.2478, CRCM434 = 0.0868, CRCM412 = 0.1511. Error bars represent standard deviation (SD) of the mean. (**D**) H&E staining of primary tumors, scale bar 400 μm. (**B**, **C**) MDA-MB-231, three independent experiments, $n = 5, 4, 4$, MDA-MB-468 one experiment $n = 6$, PDX-1432C two independent experiments, $n = 6, 4$, PDX-4295 one experiment $n = 6$, CRCM434 four independent experiments, $n = 7, 6, 4, 5$, CRCM412 four independent experiments, $n = 4, 2, 5, 5$.

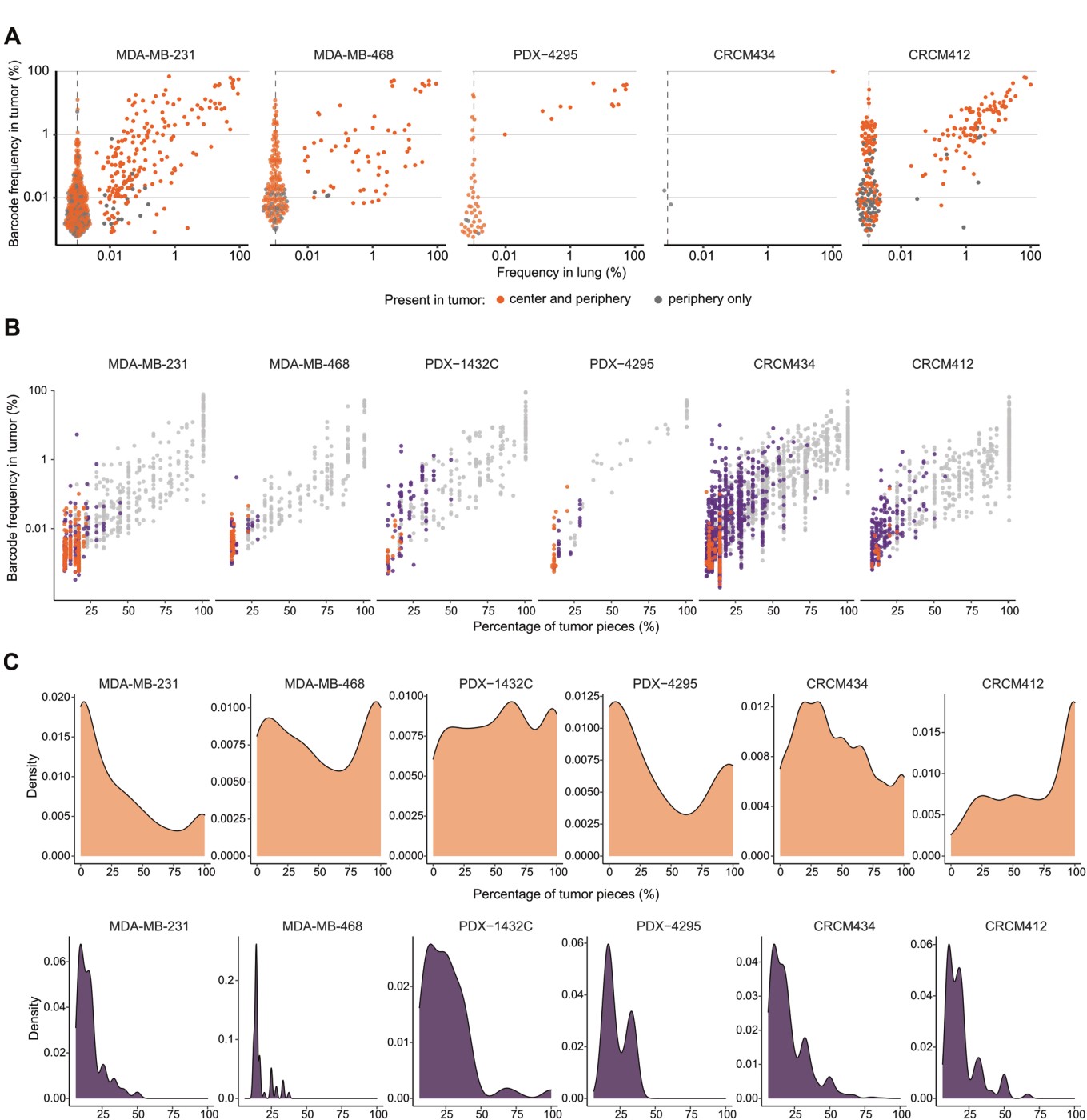

**Figure EV3.    Barcode distribution in primary tumor pieces and lungs.**

(A) Clonal relationship between barcodes detected in primary tumors and lungs. The color indicates the localization of the barcodes. If the barcodes were found in the center of the primary tumor (not exclusively), the dots are orange. If the barcodes were localized exclusively in peripheral pieces, the dots are gray. Each dot represents a barcode. Dots on dashed lines represent barcode uniquely found in tumor and not detected in lung. MDA-MB-231, three independent experiments, $n = 1, 4, 4$, MDA-MB-468 one experiment, $n = 5$, PDX-4295 two experiments, $n = 1, 2$, CRCM434 one experiment, $n = 1$, CRCM412 three independent experiments, $n = 2, 1, 4$. (B) Relationship between the frequency of individual barcodes in primary tumors and the number of pieces (in percentage) containing these barcodes. Each dot represents a barcode uniquely found in the tumor center (orange), periphery (purple) or found in center and periphery (gray). (C) Density plot showing the distribution of the number of pieces (in percentage) from the periphery containing barcodes present in the center (top), or barcodes uniquely found in peripheral pieces (bottom). (B, C) MDA-MB-231, three independent experiments, $n = 5, 4, 4$, MDA-MB-468 one experiment $n = 6$, PDX-1432C two independent experiments, $n = 6, 4$, PDX-4295 one experiment $n = 6$, CRCM434 four independent experiments, $n = 7, 6, 4, 5$, CRCM412 four independent experiments, $n = 4, 2, 5, 5$.

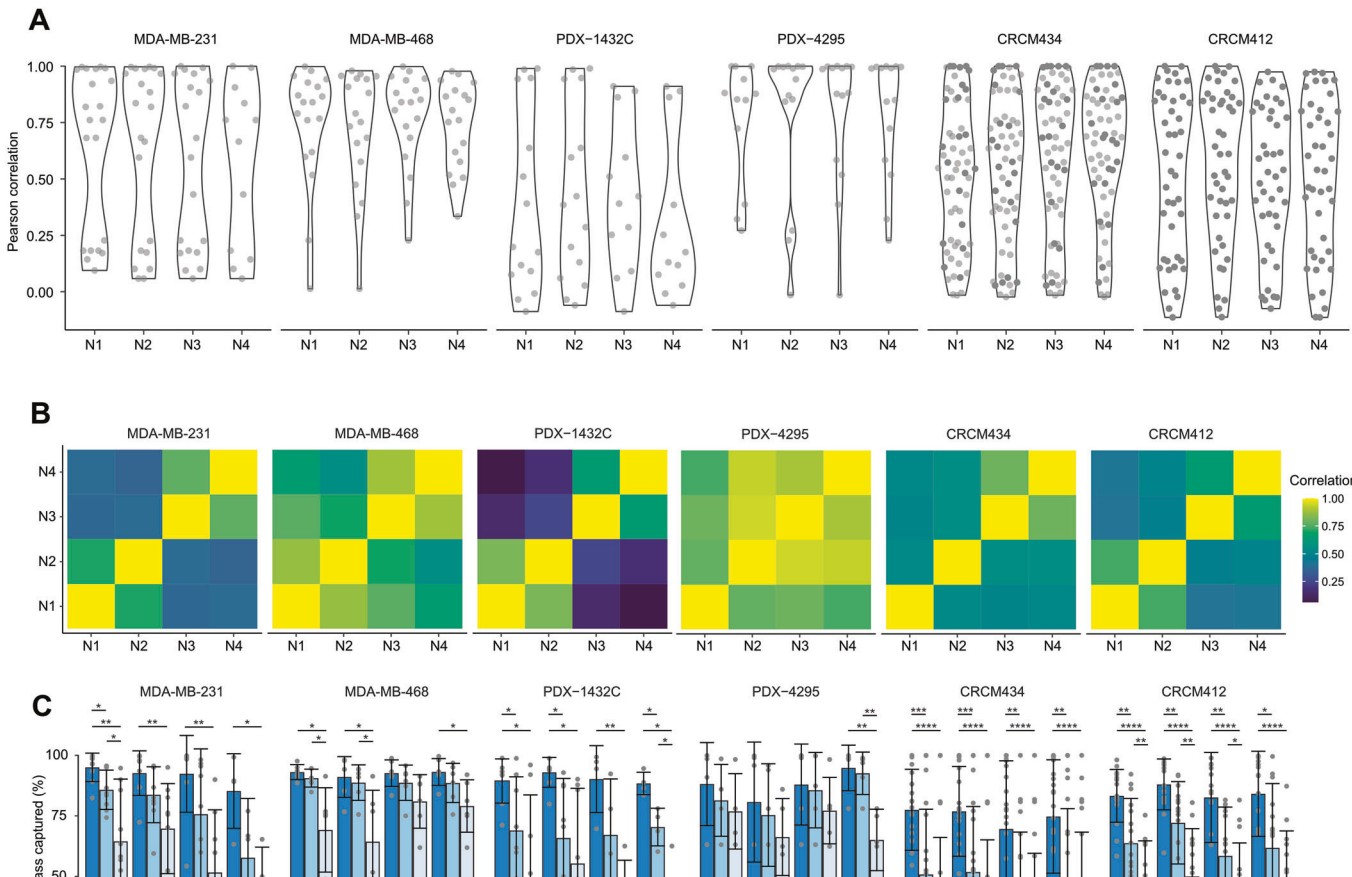

**Figure EV4. Barcode correlation in solid biopsies.**

(**A**) Correlation of clonal frequencies between needle samples and matching primary tumors. (**B**) Pearson correlation mean comparison between different needle samples from matching tumors represented as heatmap. (**C**) Percentage of primary tumor biomass captured in needle samples without threshold (left blue bar), for barcode frequency above 1% (middle bar), or above 10% (right bar). Student's unpaired *t* test, *P value < 0.05, **P value < 0.01, ***P value < 0.001, ****P value < 0.0001. Error bars represent standard deviation (SD) of the mean. Each dot corresponds to a needle biopsy. MDA-MB-231 three independent experiments, *n* = 1, 4, 4, MDA-MB-468 one experiment, *n* = 6, PDX-1432C two independent experiments, *n* = 4, 4, PDX-4295 one experiment, *n* = 5, CRCM434 four independent experiments, *n* = 7, 6, 4, 5, CRCM412 four independent experiments, *n* = 4, 2, 5, 5. P values: MDA-MB-231 N1 No threshold-1% = 0.0145, N1 No threshold-10% = 0.0072, N1 1%-10% = 0.0407, N2 No threshold-10% = 0.0064, N3 No threshold-10% = 0.0027, N4 No threshold-10% = 0.0235; MDA-MB-468 N1 No threshold-10% = 0.0195, N1 1%-10% = 0.0290, N2 No threshold-10% = 0.0260, N2 1%-10% = 0.0365, N4 No threshold-10% = 0.0237; PDX-1432C N1 No threshold-1% = 0.0383, N1 No threshold-10% = 0.0129, N2 No threshold-1% = 0.0167, N2 No threshold-10% = 0.0107, N3 No threshold-10% = 0.0051, N4 No threshold-1% = 0.0110, N4 No threshold-10% = 0.0153, N4 1%-10% = 0.0332; PDX-4295 N4 No threshold-10% = 0.0035, N4 1%-10% = 0.0051; CRCM434 N1 No threshold-1% = 0.0005, N1 No threshold-10% = 4.2e-06, N2 No threshold-1% = 0.0009, N2 No threshold-10% = 2.9e-06, N3 No threshold-1% = 0.0020, N3 No threshold-10% = 7.1e-05, N4 No threshold-1% = 0.0021, N4 No threshold-10% = 5.5e-05; CRCM412 N1 No threshold-1% = 0.0013, N1 No threshold-10% = 5.8e-07, N1 1%-10% = 0.0092, N2 No threshold-1% = 0.0040, N2 No threshold-10% = 6.4e-07, N2 1%-10% = 0.0020, N3 No threshold-1% = 0.0029, N3 No threshold-10% = 2.4e-05, N3 1%-10% = 0.0268, N4 No threshold-1% = 0.0154, N4 No threshold-10% = 5.4e-05.

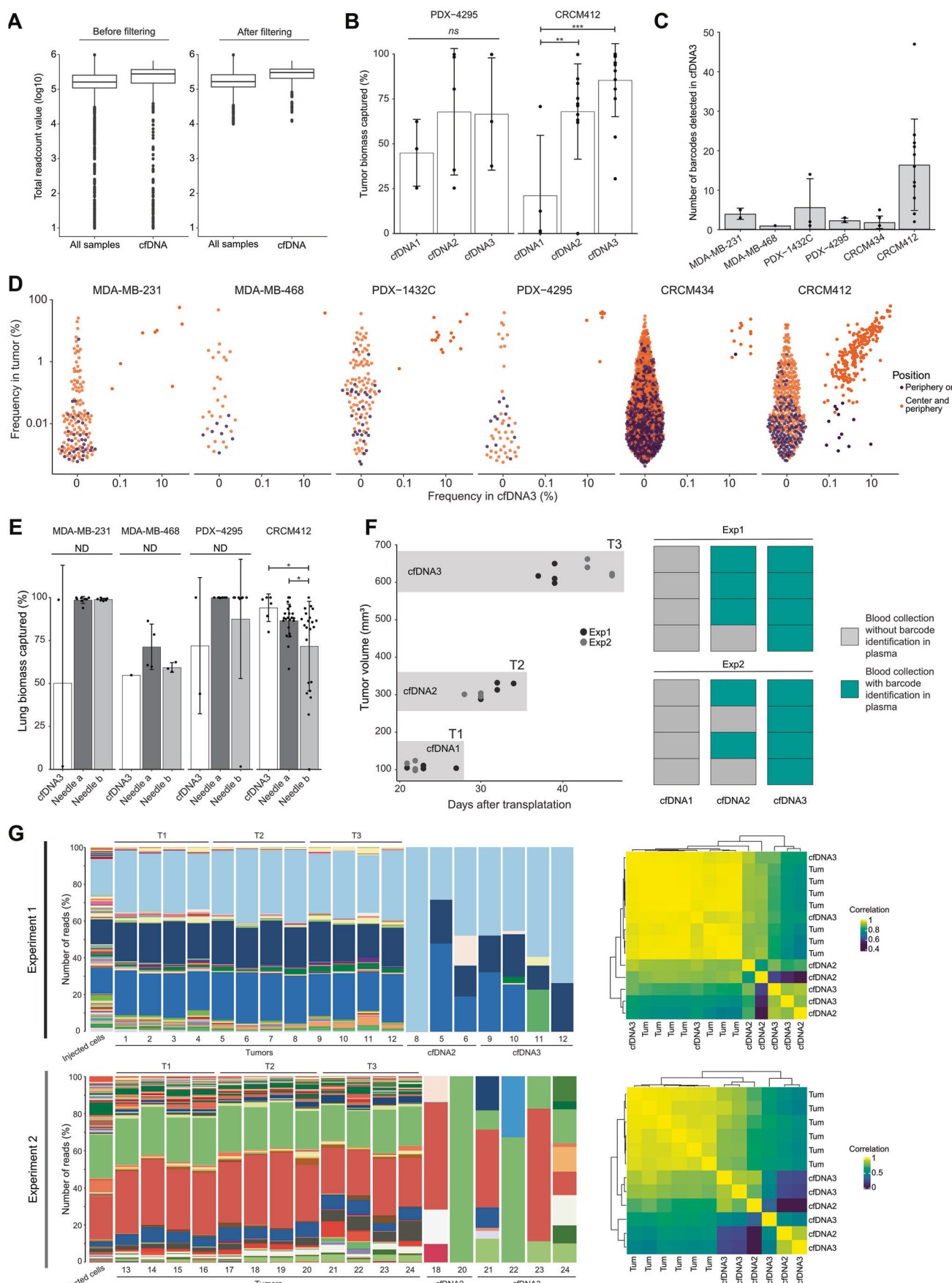

◄ **Figure EV5. Barcode analysis in cfDNA.**

(A) Barcoded sample coverage, highlighting specific read-count values for cfDNA samples versus all samples. Total read count per sample, before (left) and after filtering (right). Boxplots show the median (center line), interquartile range (box; 25th–75th percentiles), and whiskers indicate minima and maxima extending to x1.5 the interquartile range, outliers shown as individual points. (B) Percentage of tumor biomass captured at different timepoints. One-way Anova followed by Tukey multiple comparisons test, $P$ values CRCM412 cfDNA1-cfDNA2 = 0.0096, cfDNA1-cfDNA3 = 0.0003. Error bars represent standard deviation (SD) of the mean, PDX-4295 two independent experiments, $n = 1, 4$, CRCM412 four independent experiments, 4, 2, 5, 4. Each dot corresponds to a cfDNA sample. (C) Number of barcodes detected in cfDNA3 in different models. Each dot corresponds to a cfDNA sample. Error bars represent standard deviation (SD) of the mean. (D) Relationship between the barcode in primary tumor and cfDNA3. Each barcode is represented by a dot, and its color represents its location in the primary tumor. Barcodes exclusively detected in peripheral pieces are represented in purple and barcodes detected in the center are represented in orange. (C, D) MDA-MB-231 one experiment, $n = 2$, MDA-MB-468 one experiment, $n = 1$, PDX-1432C two independent experiments, $n = 1, 2$, PDX-4295 one experiment, $n = 3$, CRCM434 three independent experiments, $n = 3, 2, 2$, CRCM412 four independent experiments, $n = 4, 2, 4, 4$. (E) Lung cancer cell biomass captured with cfDNA3 samples, deep needle samples (Needle a) or shallow needle samples (Needle b). One-way Anova followed by Tukey multiple comparisons test, $P$ values: CRCM412 cfDNA1/Needle-b = 0.0486, Needle-a/Needle-b = 0.0368. Error bars represent standard deviation (SD) of the mean. Each dot corresponds to a biopsy sample. MDA-MB-231 three experiment, $n = 1, 4, 4$, MDA-MB-468 one experiment, $n = 5$, PDX-4295 two experiments, $n = 1, 2$, CRCM412 three independent experiments, $n = 2, 1, 4$. (F) Tumor volume of tumors PDX-1432C at the times of euthanasia and terminal end bleed: T1, T2 and T3 (left). Each dot corresponds to a tumor. Barcode detection in plasma at the different time points, success (green box), unsuccessful (gray box) (right panel). (G) Representation of the clonal composition in tumors from clone-splitting experiment (left panels) and their correlation between primary tumor and cfDNA (right panels). PDX-1432C two independent experiments, $n = 12, 12$. For panels (B–E) and (G), only samples in which barcodes were detected in the plasma were included.

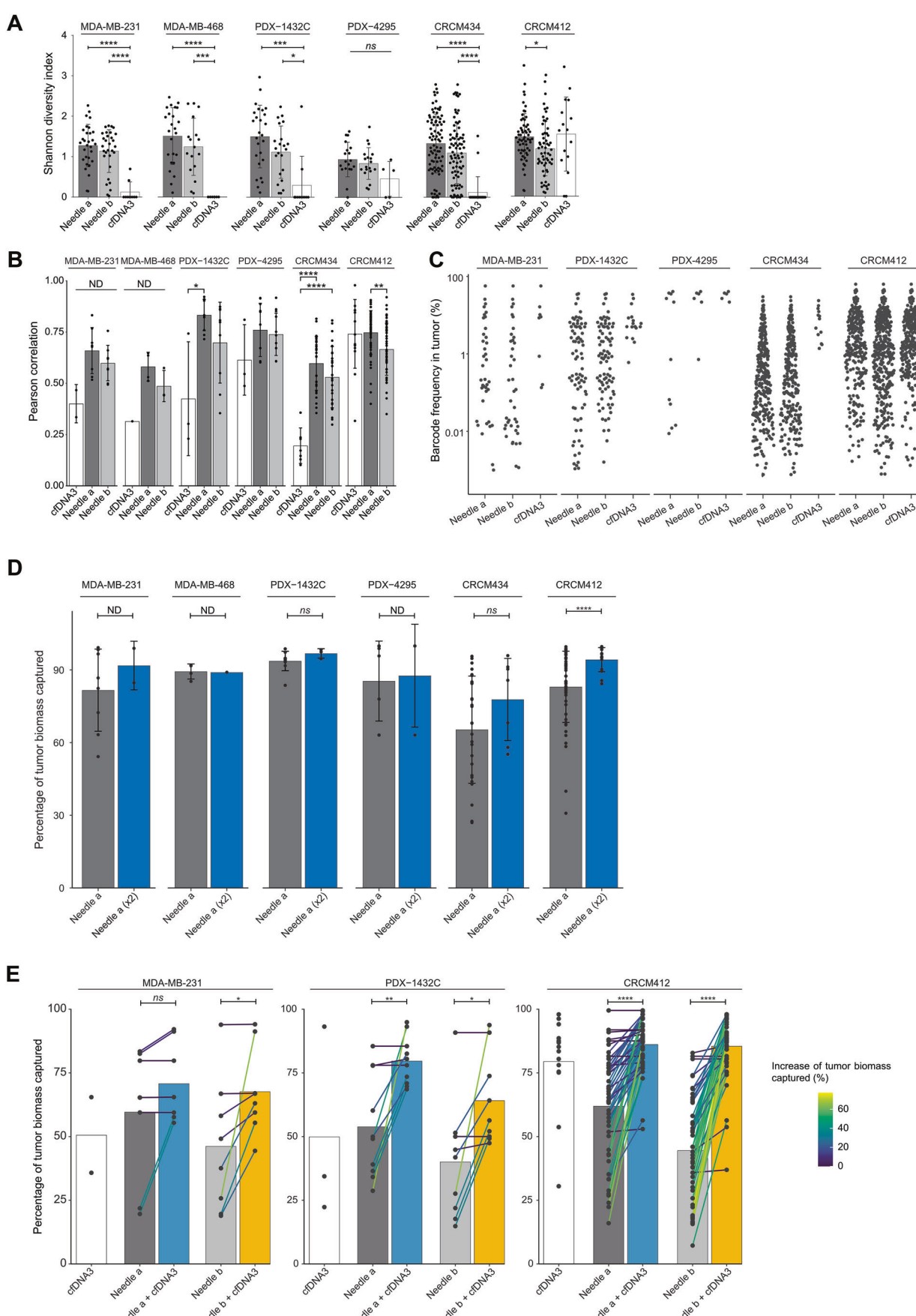

◄ **Figure EV6. Combination of solid and liquid biopsies.**

(A) Shannon diversity index from Needle and cfDNA samples. In this panel, all the cfDNA samples were included, including those with unsuccessful barcode recovery. Needle a corresponds to solid biopsy reaching the center of the tumor (deep needle sampling). Needle b corresponds to shallow sampling, covering only a quarter of the tumor diameter. Each dot corresponds to a biopsy sample. One-way ANOVA followed by Tukey multiple comparisons test, $P$ values MDA-MB-231 Needle-a/cfDNA3 = 2.4e-07, Needle-b/cfDNA3 = 4.0e-06; MDA-MB-468 Needle-a/cfDNA3 = 3.1e-05, Needle-b/cfDNA3 = 0.0008; PDX-1432C Needle-a/cfDNA3 = 0.0001, Needle-b/cfDNA3 = 0.0113; CRCM434 Needle-a/cfDNA3 = 6.5e-11, Needle-b/cfDNA3 = 1.2e-07; CRCM412 Needle-a/Needle-b = 0.0480. Error bars represent standard deviation (SD) of the mean. MDA-MB-231, three independent experiments, $n = 1,4,4$, MDA-MB-468 one experiment, $n = 6$, PDX-1432C two independent experiments, $n = 6$, 4, PDX-4295 two experiment, $n = 1$, 4, CRCM434, four independent experiments, $n = 7$, 6, 4, 5, CRCM412, four independent experiments, $n = 4$, 2, 5, 5. (B) Pearson correlation coefficients measuring the barcode similarity between biopsies and primary tumors (including all pieces). One-way ANOVA followed by Tukey multiple comparisons test, $P$ values CRCM434 cfDNA3/Needle-a = 7.9e-09, cfDNA3/Needle-b = 7.3e-07; CRCM412 Needle-a/Needle-b = 0.0032. Error bars represent standard deviation (SD) of the mean. (C) Representation of the barcode frequency in primary tumors, for barcodes detected in Needle a (deep needle samples), Needle b (shallow needle samples) and liquid biopsy (cfDNA3). Each dot represents a barcode captured in a biopsy sample. (B, C) MDA-MB-231 one experiment, $n = 2$, MDA-MB-468 one experiment, $n = 1$, PDX-1432C two independent experiments, $n = 1, 2$, PDX-4295 one experiment, $n = 3$, CRCM434 three independent experiments, $n = 3$, 2, 2, CRCM412 four independent experiments, $n = 4$, 2, 4, 4. (D) Primary tumor biomass captured in each needle sample or in two combined needle samples (blue bars). Student's unpaired $t$ test, $P$ values: PDX-1432C = 0.1069, CRCM434 = 0.3852, CRCM412 = 3.8e-05. ns: non-significant, ****$P$ value < 0.0001. Error bars represent standard deviation (SD) of the mean. MDA-MB-231, one experiment, $n = 2$, PDX-1432C two independent experiments, $n = 1, 2$, PDX-4295 two experiment, $n = 1, 2$, CRCM434, three independent experiments, $n = 3$, 2, 2, CRCM412, four independent experiments, $n = 4$, 2, 4, 4. (E) Primary tumor biomass captured with biopsy methods with barcode frequency in biopsy thresholds at > 1%. Combination of deep needle sample and cfDNA (Needle a + cfDNA3) or shallow needle sample and cfDNA (Needle b + cfDNA3). Correlated needle samples are linked with their associated increased value when cfDNA is added, and the color of the line is scaled on the increase in tumor biomass captured as percentage. Each dot corresponds to a biopsy sample. Paired $t$ test, $P$ values: MDA-MB-231 Needle-a/Needle-a+cfDN3 = 0.0836, Needle-b/Needle-b+cfDN3 = 0.0279; PDX-1432C Needle-a/Needle-a+cfDN3 = 0.0041, Needle-b/Needle-b+cfDN3 = 0.0101; CRCM412 Needle-a/Needle-a+cfDN3 = 2.0e-12 Needle-b/Needle-b+cfDN3 = 6.8e-18. ns: non-significant, *$P$ value < 0.05, **$P$ value < 0.01, ***$P$ value < 0.001, **** $P$ value < 0.0001. Error bars represent standard deviation (SD) of the mean. MDA-MB-231 one experiment, $n = 2$, PDX-1432C two independent experiments, $n = 1, 2$, CRCM412 four independent experiments, $n = 4$, 2, 4, 4. For panels (B, C) and (E), only samples in which barcodes were detected in the plasma were included.

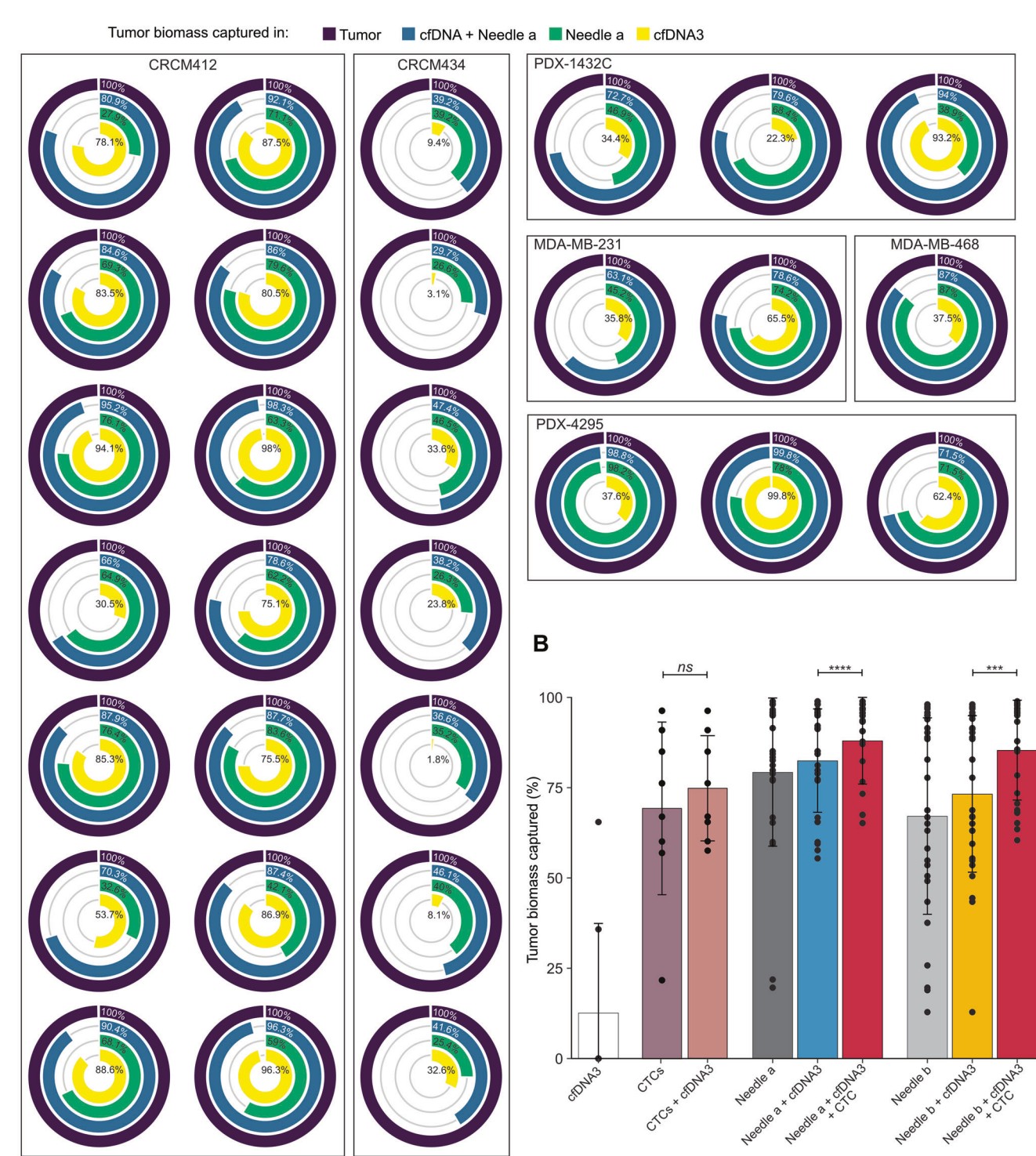

**Figure EV7.  Analysis of the tumor biomass captured in solid and liquid biopsies.**

(A) Circle plots representing the tumor biomass captured by each biopsy sampling alone or in combination, from multiple mice per models. Primary tumor plotted as reference (100%) in the outer circle (purple), inner circles represent the primary tumor biomass captured with cfDNA (yellow), deep needle samples (green) and combination of the two methods (blue). Only barcodes above 1% of the total frequency of biopsy samples were included in the computation of primary tumor biomass. MDA-MB-231 one experiment, $n = 2$, MDA-MB-468 one experiment, $n = 1$, PDX-1432C 2 independent experiments, $n = 1$, 2 PDX-4295 one experiment, $n = 3$, CRCM434 three independent experiments, $n = 3$, 2, 2, CRCM412 four independent experiments, $n = 4$, 2, 4, 4. (B) Primary tumor biomass captured with biopsy methods in MDA-MB-231, with barcodes frequency in biopsy thresholds at > 1%. Red bars indicate combinations of needles samples with cfDNA and CTCs. Paired *t* test, *P* values: CTCs/CTCs+cfDNA3 = 0.2530, Needle-a+cfDNA3/Needle-a+cfDN3 + CTCs = 2.1e-06, Needle-b+cfDNA3/Needle-b+cfDN3 + CTCs = 0.0007, ns: non-significant, ***P value < 0.001, ****P value < 0.0001. Error bars represent standard deviation (SD) of the mean. MDA-MB-231 two independent experiments, $n = 4$, 4. For these panels, only samples in which barcodes were detected in the plasma were included.

