## [Peer Review File · Molecular Systems Biology]

Genetic barcoding uncovers the clonal makeup of solid and liquid biopsies

Antonin Serrano, Tom Weber, Jean Berthelet, Sarah Ftouni, Farrah El-Saafin, Samuel Lee, Elgene Lim, Emmanuelle CHARAFFE-JAUFFRET, Christophe Ginestier, David Williams, Frederic Hollande, Belinda Yeo, Sarah-Jane Dawson, Shalin Naik, and Delphine Merino

Corresponding author(s): Delphine Merino (delphine.merino@onjcri.org.au) , Tom Weber (weber.ts@wehi.edu.au), Antonin Serrano (antonin.serrano@unimelb.edu.au)

Review Timeline:

Submission Date:	15th Jun 25
Editorial Decision:	13th Aug 25
Revision Received:	9th Nov 25
Editorial Decision:	17th Dec 25
Revision Received:	9th Jan 26
Accepted:	14th Jan 26

Editor: Jingyi Hou

Transaction Report:

13th Aug 2025

Manuscript Number: MSB-2025-13181-T

Title: Genetic barcoding uncovers the clonal makeup of solid and liquid biopsies

Author: Delphine Merino

Antonin Serrano

Tom Weber

Jean Berthelet

Sarah Ftouni

Farrah El-Saafin

Verena Wimmer

Kelly Rogers

Elgene Lim

Emmanuelle CHARAFFE-JAUFFRET

Christophe Ginestier

David Williams

Frederic Hollande

Belinda Yeo

Sarah-Jane Dawson

Shalin Naik

Dear Delphine,

Thank you for submitting your work to Molecular Systems Biology. We have now heard back from both reviewers. As you will see from the reports below, they find the study interesting and relevant. However, they raised a series of concerns, which we would ask you to address in a major revision.

I think the reviewers' recommendations are clear, so it is unnecessary to reiterate the points listed below. In particular, Reviewer #1 emphasizes the need for a clearer presentation and both reviewers have raised a few concerns about statistics.

All other issues raised by the reviewers need to be satisfactorily addressed as well. As you may already know, our editorial policy allows in principle a single round of major revision, so it is essential to provide responses to the reviewers' comments that are as complete as possible. Please feel free to contact me in case you would like to discuss in further detail any of the issues raised by the reviewers.

On a more editorial level, we would ask you to address the following issues:

- Please provide a .docx formatted version of the manuscript text (including legends for main figures, EV figures and tables). Please make sure that the changes are highlighted to be clearly visible.

- Please provide individual production quality figure files as .eps, .tif, .jpg (one file per figure).

- Please provide a .docx formatted letter INCLUDING the reviewers' reports and your detailed point-by-point responses to their comments. As part of the EMBO Press transparent editorial process, the point-by-point response is part of the Review Process File (RPF), which will be published alongside your paper.

- Please note that all corresponding authors are required to supply an ORCID ID for their name upon submission of a revised manuscript.

- We replaced Supplementary Information with Expanded View (EV) Figures and Tables that are collapsible/expandable online (see examples in <http://msb.embopress.org/content/11/6/812>). A maximum of 5 EV Figures can be typeset. EV Figures should be cited as 'Figure EV1, Figure EV2' etc... in the text and their respective legends should be included in the main text after the legends of regular figures.

Additional Tables/Datasets should be labeled and referred to as Table EV1, Dataset EV1, etc. Legends have to be provided in a separate tab in case of .xls files. Alternatively, the legend can be supplied as a separate text file (README) and zipped together with the Table/Dataset file.

For the figures and tables that you do NOT wish to display as Expanded View figures, they should be bundled together with their legends in a single PDF file called *Appendix*, which should start with a short Table of Content. Each legend should be below the corresponding Figure/Table in the Appendix. Appendix figures and tables should be referred to in the main text as:

"Appendix Figure S1, Appendix Figure S2, Appendix Table S1" etc. See detailed instructions regarding expanded view here: <https://www.embopress.org/page/journal/17444292/authorguide#expandedview>.

-Before submitting your revision, primary datasets (and computer code, where appropriate) produced in this study need to be deposited in an appropriate public database (see [http://msb.embopress.org/authorguide - dataavailability](http://msb.embopress.org/authorguide-dataavailability) <https://www.embopress.org/page/journal/17444292/authorguide#dataavailability>). Please remember to provide a reviewer password if the datasets are not yet public. The accession numbers and database should be listed in a formal "Data Availability" section (placed after Materials & Method) that follows the model below (see also <https://www.embopress.org/page/journal/17444292/authorguide#dataavailability>). Please note that the Data Availability Section is restricted to new primary data that are part of this study.

Data availability

Additional information on source data and instruction on how to label the files are available

- Our journal encourages inclusion of *data citations in the reference list* to directly cite datasets that were re-used and obtained from public databases. Data citations in the article text are distinct from normal bibliographical citations and should directly link to the database records from which the data can be accessed. In the main text, data citations are formatted as follows: "Data ref: Smith et al, 2001". In the Reference list, data citations must be labeled with "[DATASET]". A data reference must provide the database name, accession number/identifiers and a resolvable link to the landing page from which the data can be accessed at the end of the reference. Further instructions are available at .

- We updated our journal's competing interests policy in January 2022 and request authors to consider both actual and perceived competing interests. Please review the policy <https://www.embopress.org/competing-interests> and update your competing interests if necessary.

Please use the heading "Disclosure statement and competing interests".

- All Materials and Methods need to be described in the main text using our 'Structured Methods' format. According to this format, the Methods section includes a Reagents and Tools Table (listing key reagents, experimental models, software and relevant equipment and including their sources and relevant identifiers) followed by a Methods and Protocols section describing the methods, ideally using a step-by-step protocol format. The aim is to facilitate adoption of the methodologies across labs. Please download and fill our Reagents and Tools Table template (.docx), which you can find in our author guidelines: <https://www.embopress.org/page/journal/17444292/authorguide#structuredmethods>.

-Regarding data quantification:

Please ensure to specify the name of the statistical test used to generate error bars and P values, the number (n) of independent experiments (please specify technical or biological replicates) underlying each data point and the test used to calculate p-values in each figure legend. Discussion of statistical methodology can be reported in the materials and methods section, but figure legends should contain a basic description of n, P and the test applied.

Graphs must include a description of the bars and the error bars (s.d., s.e.m.).

- Please provide a "standfirst text" summarizing the study in one or two sentences (approximately 250 characters, including space), three to four "bullet points" highlighting the main findings and a "synopsis image" (550px width and 400-600 px height, PNG format) to highlight the paper on our homepage.

Here are a couple of examples:

<https://www.embopress.org/doi/10.15252/msb.20199356>

<https://www.embopress.org/doi/10.15252/msb.20209475>

<https://www.embopress.org/doi/10.15252/msb.209495>

When you resubmit your manuscript, please download our CHECKLIST (<https://www.embopress.org/pb-assets/embosite/EMBO%20Press%20Author%20Checklist-1642513524327.xlsx>) and include the completed form in your submission.

Please note that the Author Checklist will be published alongside the paper as part of the transparent process

(<https://www.embopress.org/page/journal/17444292/authorguide#transparentprocess>).

If you feel you can satisfactorily deal with these points and those listed by the referees, you may wish to submit a revised version of your manuscript. Please attach a covering letter giving details of the way in which you have handled each of the points raised by the referees. A revised manuscript will be once again subject to review and you probably understand that we can give you no guarantee at this stage that the eventual outcome will be favorable.

I look forward to receiving the revised manuscript soon.

Yours sincerely,
Jingyi

Jingyi Hou, PhD
Senior Editor
Molecular Systems Biology

We realize that it is difficult to revise to a specific deadline. In the interest of protecting the conceptual advance provided by the work, we recommend a revision within 3 months (11th Nov 2025). Please discuss the revision progress ahead of this time with the editor if you require more time to complete the revisions.

*** PLEASE NOTE *** As part of the EMBO Press transparent editorial process initiative (see our Editorial at <https://dx.doi.org/10.1038/msb.2010.72>), Molecular Systems Biology publishes online a Review Process File with each accepted manuscripts. This file will be published in conjunction with your paper and will include the anonymous referee reports, your point-by-point response and all pertinent correspondence relating to the manuscript. If you do NOT want this File to be published, please inform the editorial office at contact@molsystbiol.org within 14 days upon receipt of the present letter.

Reviewer #1:

focuses on the integration of multiple approaches in pre-clinical models, including breast cancer cell lines xenografts and patient-derived xenografts (PDXs), with the aim of exploring novel techniques to characterize tumor heterogeneity.

The novelty of the paper lies in the application of lentiviral-base genetic barcoding for studying tumor heterogeneity and tracking clonal populations. Its advantage is represented by the stable integration into genome independent of the tumor's molecular profile or temporal evolution.

The paper addresses several critical questions around ctDNA monitoring of tumour heterogeneity and has important findings.

The main finding is that ctDNA alone may not be sufficient for patient monitoring. These data are important as ctDNA has been widely viewed as a promising tool in oncology for longitudinal monitoring (enabling the study of treatment response and/or the emergence of resistance mechanisms at the genomic level) that will be a minimally invasive method to capture intratumoral heterogeneity and single and multiple metastatic sites. This work demonstrates limitations and requirements for both biopsy and ctDNA monitoring, namely;

First, when comparing cfDNA with needle biopsies (N1 and N2. and the primary tumor, there are three unpredictable scenarios: cfDNA might resemble the primary tumor more closely than either individual biopsy; cfDNA might fail to be representative because it reflects only a single clone; or cfDNA might be comparable to the biopsies.

Second, cfDNA shows its highest level of clonal representation at the third time point (cfDNA3), which suggests limitations, particularly in screening or early-stage disease, where tumor burden is minimal. It also raises the issue of temporal variability: although the lifespan of a PDX model is short, and tumor volume increases rapidly, the clones released into circulation appear to

remain largely stable over time.

Third, most of the detectable clones are dominant limiting the detection of minor clones that could be responsible for recurrence or disease progression (as explained in the discussion). A major limitation of the pre-clinical model used here may be the relatively short duration of the experiments, which do not allow for a clinically comparable follow-up of tumor evolution or of the selection dynamics that could favor minor, relapse-driving clones.

Integration of additional genomic and transcriptomic analyses, as discussed by the authors, could improve predictive power for relevant clonal populations and should be tested in improved pre-clinical models (outside the scope of this paper). The authors have developed new and potentially very useful approaches for testing methods for non-invasive measurement of tumor clonality in an experimental model in order to translate improved methods to patients.

Major points

1. Note that the manuscript is rather dense, often assumes other knowledge and will be hard for a non-expert audience to follow. Please consider editing to improve understanding. A clear list of hypotheses and aims in the introduction would help structure the MS.
2. To help general comprehension, please consider making Fig 1 into a visual abstract that summarizes the approach and models
3. Please provide further explanation in the results for the modeling of clonal density. This needs to include brief introduction to the approach and assumptions in the Results section. How were parameters tested and how did you converge on a final model?
4. Fig 1F is descriptive and it is unclear how comprehensive the representation is. Can you apply statistical methods to counts of LEGO clones in different regions to support the barcoding data?
5. Please consider framing more explicitly in the Discussion the impacts on this work on clinical studies. How might further clinical trials of biopsy methods be performed to demonstrate improved representation?

Minor points

1. In the Supplementary Figure 2a, hormone status is "not available", while for all pre-clinical models hormone status is reported as "negative". In the corresponding text is reported "We confirmed that the receptor status of the original patient tumor was conserved in the PDX, based on pathology reports". Please clarify
2. The described Figure 1g, reported in page 6, is not present in Figure 1
3. In the paragraph "The content of multiple needle biopsies from a given tumor are highly variable, but likely to contain dominant clones". The meaning of the last paragraph is unclear, and the way the concept is expressed may be confusing.
4. In Figure 3a for the first model (MDA-MB-231), cfDNA at second timepoint is not available. It needs to be clarified and specified in the text and in the figure caption. For the second model (MDA-MB-468) there seems to be no difference between the second and the third timepoint (two grey blocks and one "blue"), while the percentage in the top is 0% and 33% respectively although cfDNA has been detected in both the timepoints. Please clarify the statement.
5. In the paragraph "Barcode detection in cfDNA depends on the tumor model and tumor burden" the model is PDX-4295 not PDX-4292.

Reviewer #2:

Serrano and colleagues present a study where they use cell barcoding (both genetic and optical) to investigate intratumour heterogeneity (ITH) in breast cancer, specifically addressing the ability to estimate ITH in different types of biopsies (solid and liquid). ITH is clearly an important topic in cancer research with impacts on patients and the authors' investigations using mouse models should be commended for its direct assessment of the ability to quantify ITH. This is undoubtedly informative to the field when trying to resolve the mystery of how representative biopsies are (e.g. liquid biopsies), especially considering they use a controlled system. Furthermore, the rationale of the paper is strongly supported by computational simulations that give a theoretical foundation to their further work, which in my opinion is a powerful approach. The optical barcoding biopsies in Fig 2 are also very striking.

Upon reading the manuscript the following points arose that once addressed would allow me to recommend the article for publication.

Major comments

1. The investigation in the beginning mostly focuses on describing experimental evidence to support peripheral/boundary-driven growth in tumours. Whilst the demonstration of this with careful tumour sampling and barcoding is informative, the concept is not new and previously published works should be acknowledged and described in text, e.g. Lewinsohn et al, Nature Eco Evo, 2023; Noble et al, Nature Eco Evo, 2022; Chkhaidze et al, PLOS Computational Biology, 2019.
2. The statistical analysis of the barcodes is a little limited, simply counting the number/percentage of unique barcodes detected doesn't always give the full picture. The authors should consider complementary measures of diversity such as Shannon entropy.

3. Results from sequencing data will depend greatly on coverages (especially if they vary), this should be more explicitly addressed in text and in the manuscript, i.e. samples with low coverage/reads would detect fewer barcodes. This is especially important in the context of their ctDNA investigation.
4. It's not clear in 1d what n=13 means in the context of 3 independent experiments, is n=13 the number of mice? What here is the number of independent experiments? The authors should also consider using a mixed effects linear model to control for the fact that some tumours/mice were sampled more than others.
5. When comparing frequencies of barcodes across samples they should use a statistical test such as a chi-squared test to support their findings. As the authors allude to, the earlier ctDNA timepoints probably have very few barcodes detected, therefore their frequencies are probably explained by undersampling and not by a different shedding pattern.
6. How was the in silico modelling adapted to take into account "transplantation into the mammary fat pad"? This should be briefly elaborated on in text. In general the code of the adapted simulation should be released.
7. "To ensure the quality of the data, multiple filtering was applied to the data set". This is vague, more details are required.
8. Is it surprising that lung metastasis appear so polyclonal given an expected bottleneck before seeding?
9. Can the authors comment on the effect they think genetic drift and passaging might have on clonal diversity in their analysis?

Minor comments

1. Some statements require additional citations, e.g. "the recent development of single-cell RNA sequencing and drug prediction..." requires a citation.
2. There is no definition of what a "dominant" clone or a "minor" clone is.
3. There is no 1g as is being referred to in text
4. Arrow heads in 2a should be clearer.
5. The tumour biomass metric appears very informative as I guess it captures barcode diversity and frequency. However, it doesn't appear to be defined anywhere in the manuscript.
6. Fig 3a resolution is poor, it's hard to tell apart the mice.
7. Results in Fig 4h are repeated from Fig 4f, this should be avoided and a merged plot created.
8. "cfDNA can a good surrogate" typo.
9. Supp Fig 3, x-axis should just be "Percentage of tumour pieces".
10. Were the tumours in Fig 4C-E re-barcoded? Or do they represent the diversity of tumour establishment?
11. The authors should also release the raw FASTQ data in something like GEO.
12. What do the numbers above Supp Fig 1a mean?

We thank the reviewers for their insightful comments, that are addressed below.

Reviewer #1

focuses on the integration of multiple approaches in pre-clinical models, including breast cancer cell lines xenografts and patient-derived xenografts (PDXs), with the aim of exploring novel techniques to characterize tumor heterogeneity.

The novelty of the paper lies in the application of lentiviral-base genetic barcoding for studying tumor heterogeneity and tracking clonal populations. Its advantage is represented by the stable integration into genome independent of the tumor's molecular profile or temporal evolution.

The paper addresses several critical questions around ctDNA monitoring of tumor heterogeneity and has important findings.

The main finding is that ctDNA alone may not be sufficient for patient monitoring. These data are important as ctDNA has been widely viewed as a promising tool in oncology for longitudinal monitoring (enabling the study of treatment response and/or the emergence of resistance mechanisms at the genomic level) that will be a minimally invasive method to capture intratumoral heterogeneity and single and multiple metastatic sites. This work demonstrates limitations and requirements for both biopsy and ctDNA monitoring, namely;

First, when comparing cfDNA with needle biopsies (N1 and N2. and the primary tumor, there are three unpredictable scenarios: cfDNA might resemble the primary tumor more closely than either individual biopsy; cfDNA might fail to be representative because it reflects only a single clone; or cfDNA might be comparable to the biopsies.

Second, cfDNA shows its highest level of clonal representation at the third time point (cfDNA3), which suggests limitations, particularly in screening or early-stage disease, where tumor burden is minimal. It also raises the issue of temporal variability: although the lifespan of a PDX model is short, and tumor volume increases rapidly, the clones released into circulation appear to remain largely stable over time.

Third, most of the detectable clones are dominant limiting the detection of minor clones that could be responsible for recurrence or disease progression (as explained in the discussion). A major limitation of the pre-clinical model used here may be the relatively short duration of the experiments, which do not allow for a clinically comparable follow-up of tumor evolution or of the selection dynamics that could favor minor, relapse-driving clones.

Integration of additional genomic and transcriptomic analyses, as discussed by the authors, could improve predictive power for relevant clonal populations and should be tested in improved pre-clinical models (outside the scope of this paper). The authors have developed new and potentially very useful approaches for testing methods for non-invasive measurement of tumor clonality in an experimental model in order to translate improved methods to patients.

We thank the reviewer for acknowledging the novelty of our work and we agree that our study addresses several important questions about cfDNA. Most importantly, we have developed an approach to facilitate the study of cfDNA composition in preclinical models, which to date has been a current limitation in the field.

However, it should be recognised that the sensitivity of cfDNA detection in our study may be affected by our reliance on small DNA tags, compared to the analysis of multiple somatic mutations across large gene panels that are usually employed in clinical settings. In addition, the duration of our experiments and the need of transplantation are limitations of our study, compared to the application of ctDNA testing in the clinic. Therefore, at this stage, it is difficult to conclude that ctDNA alone may not be sufficient for patient monitoring, or inferior to solid biopsies in the clinic.

To address this, we have expanded our discussion accordingly:

“The qualitative analysis of barcodes captured in the plasma of multiple models suggest that dominant clones were represented, but clones that are under-represented in the primary tumor may not be detected. This could be due to several technical limitations. First, experiments with xenograft models are associated with a relatively short duration. This timeframe doesn’t allow for follow-up comparable to clinical settings, regarding clonal evolution and dynamics. Second, these results, which are based on the detection of genetic barcodes that are smaller than 100bp, may underestimate the representation of cfDNA fragments detected in clinical settings¹³. It is possible that some barcodes present in truncated forms were not recognized for primer annealing and therefore not detected. In clinical settings, detecting multiple gene fragments present in the plasma is likely to provide a better representation of tumor heterogeneity, provide superior detection of minor clones and improve the sensitivity of detection in cases of low tumor burden. Finally, the volume of blood and bleeding strategies used in these pre-clinical models differed from clinical settings. However, while this quantitative analysis in xenograft models, using barcode detection, might not be directly comparable to analysis of patient biopsies, it provides an opportunity to study the process of DNA shedding by cancer clones in a longitudinal manner, in the absence of therapeutic interventions.”

Major points

1. Note that the manuscript is rather dense, often assumes other knowledge and will be hard for a non-expert audience to follow. Please consider editing to improve understanding. A clear list of hypotheses and aims in the introduction would help structure the MS.

We agree that this analysis addresses a lot of questions, and we have now simplified the text and clarified several concepts, such as the definition of dominant clones and tumor biomass. We also included a list of hypotheses in the introduction:

“Here, we leveraged the use of DNA based cellular barcoding to label six human breast cancer xenograft models and explore the clonal repertoire captured in solid and liquid biopsies. By comparing the barcode repertoire present in ex-vivo needle biopsy samples and in the plasma of tumor-bearing mice, we tested the hypothesis that specific clones might be over-represented in solid biopsies, based on their random localization in primary tumors, while liquid biopsies might provide a more representative overview of clonal diversity. Furthermore, we explored the factors influencing the detection of barcodes in solid and liquid biopsies, such as tumor burden, tumor necrosis and the intrinsic properties of the cancer cells. This analysis provided a unique opportunity to comprehensively assess the ability of different sampling methods to capture ITH.”

2. To help general comprehension, please consider making Fig 1 into a visual abstract that summarizes the approach and models

We included a graphical abstract and a summary as part of the Synopsis section, to summarize the experimental design, models and readout used in our study, and to enhance general comprehension at the start of the manuscript.

Synopsis: In this study, genetic barcoding was used to label breast cancer cells, track their fate in vivo, and study the level of intra-tumoral heterogeneity captured in solid and liquid biopsies. The analysis revealed that:

- Genetic tags can be detected in the plasma of pre-clinical barcoded models;
- DNA shedding can significantly varies depending on the model;
- Combining cfDNA and solid biopsies is likely to provide a better representation of intra-tumoral heterogeneity.

3. Please provide further explanation in the results for the modeling of clonal density. This needs to include brief introduction to the approach and assumptions in the Results section. How were parameters tested and how did you converge on a final model?

The results section regarding the modelling has been clarified as followed:

“To better understand the spatial distribution of the clones in primary tumors and how it affects the diversity captured in solid biopsies, a previously established simulation model was adapted⁴¹ (Fig. 1a). This stochastic agent-based model predicts the 3D growth of cancer cells based on three cellular parameters: birth rate, death rate and mobility. Mimicking the number of clones detected in previous cellular barcoding experiments⁹, simulations of fat pad transplantations experiments were initiated in silico with a starting number of two hundred cells, where each cell was given a unique identity (using a virtual tag stored in the genotype slot), thereafter inherited by their respective progeny. No other change was made to the model, previously detailed in ⁴¹. As expected, the resulting simulation of clonal growth confirmed the clonal ‘patchiness’ of the tumors previously observed using genetic barcoding⁹. However, the analysis of the spatial distribution of the clones in 3D revealed that clonal density in the center of the tumor was higher compared to the periphery (Fig 1b). Indeed, the virtual dissection of

these tumors into five pieces, and quantification of number of clones in each, indicated that the center of the tumor (piece E) exhibited a significantly increased number of clones.”

4. Fig 1F is descriptive and it is unclear how comprehensive the representation is. Can you apply statistical methods to counts of LEGO clones in different regions to support the barcoding data?

Fig. 1F was descriptive, as only one LeGO image per model (MDA-MB-231 and PDX CRCM 434) was collected for this study, to provide a visual representation of the quantitative analysis provided by the genetic barcoding analysis. While the MDA-MB-231 image clearly indicates that clonal diversity is increased in the center of the tumor (Fig. 1f of the original manuscript), this observation is not obvious in the PDX tumor (Fig. 2e of the original manuscript). The small number of barcodes used with optical barcoding and the depth of the sections, compared to the number of barcodes and the size of the pieces analysed with genetic barcodes, may imply that we will need multiple images per tumor, multiple tumors and models to provide a robust quantitative analysis of this system and apply statistical methods.

To do so, we will need to barcode multiple models with 31 colour LeGO system, amplify individual clones, transplant in multiple mice per model, and generate a new pipeline of analysis to provide a robust quantification of tumor heterogeneity. As this would likely take several months and significantly delay publication, without offering meaningful addition insight into the representativity of tumor biopsies, we decided to remove the LeGO part from this study. We are hoping to investigate this in more depth in the future, dedicating a study to the analysis of the spatial distribution and characterisation of the clones in optically barcoded models.

5. Please consider framing more explicitly in the Discussion the impacts on this work on clinical studies. How might further clinical trials of biopsy methods be performed to demonstrate improved representation?

To emphasise the impact of this work in the discussion, we highlighted two aspects:

- *“Additional studies linking genomic analysis and clonal information in these models will be required to better understand the process of DNA shedding, and why some models, such as PDX CRCM412, shed a significant amount of cfDNA, while others, such as MDA-MB-231, don’t, despite being highly metastatic. Such investigations will be required to optimize cfDNA use in disease monitoring, for instance via the identification of biomarkers associated with false negatives, by characterizing aggressive clones that do not shed cfDNA in the plasma.”*
- *“... sequencing analysis of 351 samples from patients with diverse cancer types suggested that the combination of both solid and liquid biopsies offers a more therapeutically valuable representation of tumor heterogeneity in clinical settings²³. Similar studies, comparing the utility of both liquid and solid biopsies in clinical settings, will be required to optimize their use, not only in the context of tumor heterogeneity, but also to identify the risks of disease progression and drug resistance.”*

Reviewer #1 - Minor points

1. In the Supplementary Figure 2a, hormone status is "not available", while for all pre-clinical models hormone status is reported as "negative". In the corresponding text is reported "We confirmed that the receptor status of the original patient tumor was conserved in the PDX, based on pathology reports". Please clarify

We apologise for the confusion. The tumors from patients 1432C, CRCM412, CRCM434 and 4295 were triple negative, but the necrosis status of tumors CRCM412, CRCM434 and 4295 is unknown. This has been clarified in the legend.

2. The described Figure 1g, reported in page 6, is not present in Figure 1

This was a mistake, the text was referring to Fig. 1f. This panel has now been removed from the manuscript, as discussed above.

3. In the paragraph "The content of multiple needle biopsies from a given tumor are highly variable, but likely to contain dominant clones". The meaning of the last paragraph is unclear, and the way the concept is expressed may be confusing.

We rephrased this paragraph, followed by one short summary sentence (Page 7 of the revised manuscript):

"In order to determine if solid biopsies were, overall, representative of the ITH of primary tumors, we determine the percentage of reads from the primary tumors that were covered by barcodes detected in needles. We found that clones detected in needle biopsies represented 80 to 90% of the tumor biomass (referring here to the total number of reads in the whole tumor), regardless of their orientation (Fig. EV4c).

Overall, these results demonstrated that, although the content of needle biopsies may vary depending on their orientation and dominant clones might be missed during sampling, these biopsies are largely representative of the primary tumor biomass."

4. In Figure 3a for the first model (MDA-MB-231), cfDNA at second timepoint is not available. It needs to be clarified and specified in the text and in the figure caption. For the second model (MDA-MB-468) there seems to be no difference between the second and the third timepoint (two grey blocks and one "blue"), while the percentage in the top is 0% and 33% respectively although cfDNA has been detected in both the timepoints. Please clarify the statement.

The percentage above the second timepoint for the MDA-MB-468 was mistakenly labelled as 0%. This has been corrected to 1/3. To avoid ambiguity, we have revised the figure to report the values as "X/N mice with successful detection"

5. In the paragraph "Barcode detection in cfDNA depends on the tumor model and tumor burden" the model is PDX-4295 not PDX-4292.

This mistake has been corrected in the revised version of the manuscript.

Reviewer #2 - Major comments

Serrano and colleagues present a study where they use cell barcoding (both genetic and optical) to investigate intratumor heterogeneity (ITH) in breast cancer, specifically addressing the ability to estimate ITH in different types of biopsies (solid and liquid). ITH is clearly a important topic in cancer research with impacts on patients and the authors investigations using mouse models should be commended for its direct assessment of the ability to quantify ITH. This is undoubtedly informative to the field when trying to resolve the mystery of how representative biopsies are (e.g. liquid biopsies), especially considering they use a controlled system. Furthermore, the rationale of the paper is strongly supported by computational

simulations that give a theoretical foundation to their further work, which in my opinion is a powerful approach. The optical barcoding biopsies in Fig 2 are also very striking.

We thank the review for their encouraging comments. We agree that this strategy offers the opportunity to study clonal dynamics in a controlled system, *which could be used for multiple preclinical studies. This point has been added in the discussion: “the main advantage of this strategy is that it enables to study the fate of labelled cells and their progeny in primary tumors and subsequent sampling in an unbiased, controlled and comprehensive manner, using multiple mice per cohort.”*

Upon reading the manuscript the following points arose that once addressed would allow me to recommend the article for publication.

1. The investigation in the beginning mostly focuses on describing experimental evidence to support peripheral/boundary-driven growth in tumors. Whilst the demonstration of this with careful tumor sampling and barcoding is informative, the concept is not new and previously published works should be acknowledged and described in text, e.g. Lewinsohn et al, Nature Eco Evo, 2023; Noble et al, Nature Eco Evo, 2022; Chkhaidze et al, PLOS Computational Biology, 2019.

These studies have been added to the revised version, in the results:

“Altogether, in silico modelling and empirical results based on barcoding analysis converged towards the conclusion that in preclinical models, the center of tumors is higher in clonal density compared to the periphery. This observation supports previous studies using the LeGO technology⁴³, sequencing and modeling analyses (e.g. ⁴⁴⁻⁴⁶), which have demonstrated that clones from the outer region of tumors are likely to have a higher growth rate compared to clones present in the center.”

And in the discussion:

“In the clinic, needles are often oriented towards the center of a tumor. Indeed, both computational modeling and empirical data from cellular barcoding experiments highlighted that the center of non-necrotic tumors is significantly enriched in barcodes compared to the periphery. While a similar observation has been made in various cancer types, using genomic analysis and modeling⁴³⁻⁴⁶, it would be important to confirm this in immunocompetent preclinical models, in models that are not relying on tumor transplantation, and in patient samples.”

2. The statistical analysis of the barcodes is a little limited, simply counting the number/percentage of unique barcodes detected doesn't always give the full picture. The authors should consider complementary measures of diversity such as Shannon entropy.

We have incorporated the analysis of the Shannon diversity index as a complementary measure of barcode diversity. New panels have been added: Fig. EV1b, Fig. EV2c, Fig. EV6a. We have included corresponding descriptions and interpretations in the main text to provide a more comprehensive analysis of barcode diversity.

3. Results from sequencing data will depend greatly on coverages (especially if they vary), this should be more explicitly addressed in text and in the manuscript, i.e. samples with low coverage/reads would detect fewer barcodes. This is especially important in the context of their ctDNA investigation.

We agree that sequencing depth can play a role in barcode detection. To minimise this effect, barcoded samples underwent a two-step amplification process: an initial barcode specific amplification, followed by the addition of sequencing adapters. This strategy helps mitigate the impact of low input material by selectively amplifying barcodes. In addition, to minimise the potential bias coming from low-coverage samples, we applied a filtering step that excludes any samples with fewer than 10,000 total reads.

Panel above added in Fig. EV5a: Barcoded sample coverage, highlighting specific read count values for cfDNA samples versus all samples. Total read count per sample, before (left) and after filtering (right).

4. It's not clear in 1d what $n=13$ means in the context of 3 independent experiments, is $n=13$ the number of mice? What here is the number of independent experiments? The authors should also consider using a mixed effects linear model to control for the fact that some tumors/mice were sampled more than others.

Indeed, the $n=13$ referred to the total number of mice across the 3 independent experiments. The number of mice per experiment is respectively 5, 4, 4. The full details of number of mice per experiment per figure panel has been generated as an extended file (Table EV1), and the numbers of mice and experiments have been clarified in the legends.

We also incorporated a linear mixed-effects model with mouse ID as a random intercept to control for the unequal number of samples collected per tumor, ensuring that mice with more observations were not overweighted. The panel below was including in Fig. EV2b.

Panel above added in Fig. EV2b: Mixed-effect model analysis of barcode distribution across tumor locations and models. Each color represents a specific mouse, each dot corresponds to a tumor piece and the line represents the linear interpolation between average for each tumor. Pairwise contrasts between “Center” and “Periphery” locations within each tumor model, based on the estimated marginal means derived from the mixed-effects model, * = p-value <0.05, **= p-value <0.01, *** = p-value <0.001, ****= p-value <0.0001. MDA-MB-231, three independent experiments, n=5, 4, 4, MDA-MB-468 one experiment n=6, PDX-1432C two independent experiments, n=6, 4, PDX-4295 one experiment n=6, CRCM434 four independent experiments, n=7,6,4,5, CRCM412 four independent experiments, n=4, 2, 5, 5.

5. When comparing frequencies of barcodes across samples they should use a statistical test such as a chi-squared test to support their findings. As the authors allude to, the earlier ctDNA timepoints probably have very few barcodes detected, therefore their frequencies are probably explained by undersampling and not by a different shedding pattern.

We thank the reviewer for the suggestion, the percentage in Fig. 3a refers the proportion of mice with successful barcode detection at each timepoint, not barcode frequencies. Because these data represent counts of mice with/without detection (e.g., 2 out of 6), they do not represent barcode frequency distributions across samples. To avoid ambiguity, we have revised the figure to report the values as “X/N mice with successful detection,” which more accurately reflects the underlying data and clarifies that this analysis does not imply different shedding frequencies.

6. How was the in silico modelling adapted to take into account "transplantation into the mammary fat pad"? This should be briefly elaborated on in text. In general the code of the adapted simulation should be released.

The results section regarding the modelling has been clarified as followed:

“This stochastic agent-based model predicts the 3D growth of cancer cells based on three cellular parameters: birth rate, death rate and mobility. Mimicking the number of clones detected in previous cellular barcoding experiments⁹, simulations of fat pad transplantations experiments were initiated in silico with a starting number of two hundred cells, where each cell was given a unique identity (using a virtual tag stored in the genotype slot), thereafter inherited by their respective progeny. No other change was made to the model, previously detailed in ⁴¹.”

The code for the simulation has been made available with the codes used to generate all figures (see data availability section).

7. "To ensure the quality of the data, multiple filtering was applied to the data set". This is vague, more details are required.

The revised version of the manuscript provides more detailed information about the filtering steps applied. The new method section includes the following text:

"To ensure the quality of the data, the dataset was filtered as follow. First barcode read counts less than or equal to 10 were set to zero within each sample. Next, samples with fewer than 10000 total reads were excluded from further analysis. Then replicate and quintuplicate samples with a Pearson correlation inferior to 0.6 were removed from the analysis (except for cfDNA samples). Finally, barcodes present in less than 2 replicates were discarded. Replicates were then pooled by adding read count values of the barcode and normalized. The value of each barcode within individual tumor pieces were cumulatively added and normalized to recreate the profile of the full tumor as displayed in Fig. 2C. Finally, barcodes only present in primary tumors were conserved across samples. All subsequent visualization and statistical analysis were performed in R. When $n < 3$, statistical analyses were not determined (ND)."

8. Is it surprising that lung metastasis appear so polyclonal given an expected bottleneck before seeding?

Fig. EV2 indicates the frequency of clones present in the lungs (on the y axis) and in the primary tumors (on the y axis). Most clones are on the dashed lines, i.e. exclusively found in the tumors and not detected in the lungs. In average, 29.6 to 41.5% of barcodes present in the primary tumors are detected in the lungs (Rebuttal Fig. 1, below). These results are in agreement with previous studies that have shown that the metastatic process is highly selective. However, the primary tumors were not resected, so it is unclear whether these clones have any 'seeding' potential.

Rebuttal Fig. 1: Percentage of barcodes detected in the lungs, compared to primary tumors. Each dot corresponds to the percentage of barcodes for a given mouse, in each model. The percentage on top of the graph represents the mean for each model, and the error bars represent standard variations (SD).

9. Can the authors comment on the effect they think genetic drift and passaging might have on clonal diversity in their analysis?

Genetic drift has been described in vitro and in vivo, in response to time and passaging. It is highly possible that our study, focussing on barcode analysis (i.e. the fate of labelled cancer cells and their progeny) rather than genomic heterogeneity, the extent of diversity captured in primary tumors and biopsies is under-estimated.

This paragraph has been added to the discussion:

"It is likely that barcoded clones were subject to genetic drifts over time and passaging, as previously demonstrated with cell lines and PDXs, both in vitro ⁴⁸ and in vivo ³⁰. By focusing on barcode analysis rather than genomic heterogeneity, this study may under-estimate the extent of diversity captured in primary tumors and biopsies. It may also highlight a level of heterogeneity that is not clinically relevant to guide therapeutic decisions. However, the main advantage of this strategy is that it enables to study the fate of labelled cells and their progeny in primary tumors and subsequent sampling in an unbiased, controlled and comprehensive manner, using multiple mice per cohort. In this context, future studies integrating barcode detection and genomic analysis will provide additional insights into clonal evolution, and its impact on the extent of genomic diversity captured in biopsies. These models could also be used to improve methods to detect tumor material in liquid biopsies (based on cfDNA, methylation, CTCs) and to assess the impact of specific therapies on DNA shedding."

Minor comments

1. Some statements require additional citations, e.g. "the recent development of single-cell RNA sequencing and drug prediction..." requires a citation.

We added a recent article and a review to illustrate this statement:

"Furthermore, the recent development of single-cell RNA sequencing (scRNA-seq) analysis and drug prediction from transcriptional signatures, rather than mutational status, highlights the utility of capturing and analyzing intact cells in cancer research and precision medicine (e.g. ^{27,28})."

2. There is no definition of what a "dominant" clone or a "minor" clone is.

In our study, we considered a dominant clone to represent a significant proportion of the tumor biomass and included a cut-off of 1%, in opposition to minor clones, that have a frequency < 1%. This has been clarified in the text:

"...dominant clones (here defined as a clone with a read frequency greater than 1% of the total reads of the tumor) were present in a larger number of pieces compared to minor clones, and the barcodes present exclusively in the center or the periphery were minor clones (Fig. EV3b)."

3. There is no 1g as is being referred to in text

The text was referring to Fig. 1f, which has been removed in the new version of the manuscript.

4. Arrow heads in 2a should be clearer.

The size of the arrow heads has been increased in Fig. 2a.

5. The tumor biomass metric appears very informative as I guess it captures barcode diversity and frequency. However, it doesn't appear to be defined anywhere in the manuscript.

The tumor biomass usually refers to the mass or volume of a tumor. In the context of our barcoding analysis, the biomass represents to total number of reads in the whole tumor.

We added a definition of the tumor biomass when we first mention it, page 7 *"We found that clones detected in needle biopsies represented 80 to 90% of the tumor biomass (referring here to the total number of reads in the whole tumor), regardless of their orientation (Fig. EV4c)."*

6. Fig 3a resolution is poor, it's hard to tell apart the mice.

A new version of the figures has been provided, with a higher resolution.

7. Results in Fig 4h are repeated from Fig 4f, this should be avoided and a merged plot created.

A merged plot was created and included in Fig. 4f of the revised manuscript. The new panel shows the percentage of tumor biomass captured by:
cfDNA3, needle A, cfDNA3 + needle a, needle b, cfDNA3 + needle b

8. "cfDNA can a good surrogate" typo.

This typo has been corrected in the revised manuscript.

9. Supp Fig 3, x-axis should just be "Percentage of tumor pieces".

This has been modified in the revised version.

10. Were the tumors in Fig 4C-E re-barcoded? Or do they represent the diversity of tumor establishment?

We agree that the text describing these panels was misleading. The tumors from Fig. 4c-e were representative examples of data from 3 different mice included in Fig. 4b. The graphs represent the diversity of primary tumors and biopsies (cfDNA3, N1a and N2a) in 3 animals showing different patterns.

We clarified this in the text:

"It is important to note that in these pre-clinical models, the barcode content of biopsies varied significantly, not only between models, but also between mice from the same model, regardless of the method. This could be due to the low detection rate and resulting 'randomness' obtained when sampling small amount of tissues and limited blood volumes. Indeed, comparing the representativity of barcodes captured by needle and liquid biopsies in individual mice from the same cohort, three distinct patterns were identified. In the first pattern (e.g. Fig. 4c as a representative example), the barcode repertoire from each biopsy sample was unique, as needle and liquid biopsies contained different sets of barcodes but the cfDNA barcode repertoire better recapitulated tumor heterogeneity. In the second pattern (Fig. 4d), needle and liquid biopsies also showed different sets of barcodes, but solid biopsies were a

better surrogate of primary tumor ITH, while cfDNA only captured one clone from the primary tumor. The last pattern (Fig. 4e) showed similarity between needle and cfDNA biopsies, where both biopsy methods accurately represented the primary tumor.”

11. The authors should also release the raw FASTQ data in something like GEO.

This data is now available.

12. What do the numbers above Supp Fig 1a mean?

The numbers on top of the bars represents the average number of barcodes per model. This has been clarified in the legend.

We also noticed that the CTC control and combination of CTC and cfDNA were missing in Supp Fig. 7b, and these results have now been included in the revised version of the manuscript.

17th Dec 2025

Manuscript Number: MSB-2025-13181R

Title: Genetic barcoding uncovers the clonal makeup of solid and liquid biopsies

Author: Antonin Serrano

Tom Weber

Jean Berthelet

Sarah Ftouni

Farrah El-Saafin

Samuel Lee

Elgene Lim

Emmanuelle CHARAFFE-JAUFFRET

Christophe Ginestier

David Williams

Frederic Hollande

Belinda Yeo

Sarah-Jane Dawson

Shalin Naik

Delphine Merino

Dear Prof Merino,

Thank you for sending us your revised manuscript. We have now heard back from Reviewer#2 who was asked to re-evaluate your study. As you will see, the reviewer is overall satisfied with the revisions made. Before we can formally accept your manuscript for publication, we would ask you to address the following issues:

1. the remaining minor comments from Reviewer #2.

On a more editorial level:

1. Please include up to five keywords in the manuscript file.

2. Remove "Authors' contribution" section from the manuscript file.

3. Funding information:

- Please ensure that the funding information in the manuscript text matches the details entered in the submission system.
- The following funding sources are missing in the submission system and need to be added: Melbourne Research Scholarship, NBCF (Investigator Initiated Research Grant IIRS0049)
- Grant numbers are missing in the submission system for the following funders: Victorian Cancer Agency (MCRF21011), NHMRC (GNT2012196 and GNT2027459)
- Funding information included in the Comments box could not be extracted. Therefore, all funders should be added explicitly to the "More Funders" list in the submission system.

4. Please provide a "standfirst text" summarizing the study in one or two sentences (approximately 250 characters, including space), three to four "bullet points" highlighting the main findings.

5. "Conflict of interest" should be renamed to "disclosure and competing interests statement".

6. The references need to be formatted according to the Molecular Systems Biology reference style. Please list up to 10 co-authors of a paper before adding et al. in the reference list. Citations should be listed in alphabetical order.

7. "Data and code Availability" should be renamed to "Data availability". Please remove the reviewer access link and make sure the datasets are publicly available upon the acceptance of the manuscript.

8. Since Tables EV1 and EV2 are quite complex, they should be converted to Dataset EV1 and Dataset EV2. Source file names, titles, legends, and manuscript callouts should be updated accordingly. Legends should be provided in a separate tab/sheet within each Excel file.

9. Source Data

- Since Figures 1F and 2E have been removed from the revised manuscript, please untick these panels in the source data

checklist.

- Source data should be organized following a one figure/one folder scheme and uploaded as .zip files. For example, all source data files for Figure 1 should be saved in a single folder, zipped, and uploaded as "SD Figure 1.zip."

10. Please address the following issues in figure legends:

- Please note that the figure EV-2d is mislabeled as figure EV-2c in the manuscript. This needs to be rectified.
- Please note that the exact p values are not provided in the legends of figures 1d,e; 4b,f; EV-1b; EV-4c; EV-5b,e; EV-6a,b,d,e; EV-7b.
- Please indicate the statistical test used for data analysis in the legends of figures EV-2b; EV-6a.
- Please note that the box plots need to be defined in terms of minima, maxima, centre, bounds of box and whiskers, and percentile in the legends of figure EV-5a.
- Please note that the error bars are not defined in the legends of figures EV-2c; EV-5c; EV-6a.
- Please note that the measure of center for the error bars needs to be defined in the legends of figures 1e; 4b,f; EV-4c; EV-5b,e; EV-6b,d,e; EV-7b.
- Please note that for heatmap present in figure EV-1e a numbered scale bar is not provided. This needs to be rectified.

11. Sections need to be named and the order should be corrected: Title page - Abstract - Keywords - Introduction - Results - Discussion - Methods - Data Availability - Acknowledgements - Disclosure and Competing Interests Statement - References - Figure Legends - Table(s) - Expanded View Figure Legends.

Kind regards,
Jingyi

Jingyi Hou, PhD
Senior Editor
Molecular Systems Biology

*** PLEASE NOTE *** As part of the EMBO Press transparent editorial process initiative (see our Editorial at <https://dx.doi.org/10.1038/msb.2010.72> , Molecular Systems Biology will publish online a Review Process File to accompany accepted manuscripts. When preparing your letter of response, please be aware that in the event of acceptance, your cover letter/point-by-point document will be included as part of this File, which will be available to the scientific community. More information about this initiative is available in our Instructions to Authors. If you have any questions about this initiative, please contact the editorial office (msb@embo.org).

Reviewer #2:

Overall, I am very happy with the response to my comments. The authors have taken a positive and open-minded approach to addressing the points and have addressed them appropriately. I have some minor additional comments below, but I am happy to recommend for acceptance. I congratulate the authors on a comprehensive manuscript.

Minor comments

- Proofread the text more thoroughly, phrasing is slightly off, e.g. "subject to genetic drifts", typically this is just "genetic drift", another example, "it enables to study the fate"
- "However, when considering the Shannon diversity index, a measure that reflects both barcode richness and evenness, all

PDXs, except PDX-1432C, showed a higher diversity." - do they mean PDX-495?

- Regarding my chi-squared test comment, I was referring to barcode frequencies as in Fig 3b. Apologies for my lack of clarity. To me the lack of diversity in cfDNA1 looked like it may be due to undersampling barcodes. However, the previous comment addressing coverage concerns helps to address this as thousands of barcode observations are likely to produce high significance in this test (and make me realise the test is probably not appropriate).

- Regarding my lung question, I realise on second review that the whole lung was sequenced and not individual lung lesions. Therefore, the barcodes detected could represent lots of micro-metastases. I would expect individual lung masses to be made of a handful of seeding cells and therefore have low diversity. Perhaps this is worth clarifying in the final manuscript? i.e. that it is whole lung.

We thank Reviewer #2 for their time, positive comments and helpful suggestions to improve this manuscript.

Reviewer #2:

Overall, I am very happy with the response to my comments. The authors have taken a positive and open-minded approach to addressing the points and have addressed them appropriately. I have some minor additional comments below, but I am happy to recommend for acceptance. I congratulate the authors on a comprehensive manuscript.

Minor comments

- Proofread the text more thoroughly, phrasing is slightly off, e.g. "subject to genetic drifts", typically this is just "genetic drift", another example, "it enables to study the fate". These mistakes have been corrected in the revised version, and the text has been thoroughly proofread.

- "However, when considering the Shannon diversity index, a measure that reflects both barcode richness and evenness, all PDXs, except PDX-1432C, showed a higher diversity." - do they mean PDX-495? This was a mistake that has been corrected in the revised version.

- Regarding my chi-squared test comment, I was referring to barcode frequencies as in Fig 3b. Apologies for my lack of clarity. To me the lack of diversity in cfDNA1 looked like it may be due to undersampling barcodes. However, the previous comment addressing coverage concerns helps to address this as thousands of barcode observations are likely to produce high significance in this test (and make me realise the test is probably not appropriate). Indeed, adding Fig. EV5a clarified in the revised version that the differences observed in cfDNA1 were not due to a low coverage of barcode sequencing.

- Regarding my lung question, I realise on second review that the whole lung was sequenced and not individual lung lesions. Therefore, the barcodes detected could represent lots of micro-metastases. I would expect individual lung masses to be made of a handful of seeding cells and therefore have low diversity. Perhaps this is worth clarifying in the final manuscript? i.e. that it is whole lung. This is a valid point, which has now been clarified in the Results and Methods sections.

14th Jan 2026

Manuscript number: MSB-2025-13181RR

Title: Genetic barcoding uncovers the clonal makeup of solid and liquid biopsies

Dear Prof Merino,

Thank you again for sending us your revised manuscript. We are now satisfied with the modifications made and I am pleased to inform you that your paper has been accepted for publication.

You may qualify for financial assistance for your publication charges - either via a Springer Nature fully open access agreement or an EMBO initiative. Check your eligibility: <https://link.springer.com/journal/44320/how-to-publish-with-us>

Sincerely,
Jingyi

Jingyi Hou, PhD
Senior Editor
Molecular Systems Biology

>>> Please note that it is Molecular Systems Biology policy for the transcript of the editorial process (containing referee reports and your response letter) to be published as an online supplement to each paper. If you do NOT want this, you will need to inform the Editorial Office via email immediately. More information is available here: <https://link.springer.com/partners/embo-press/editorial-policies#Peer%20review>